# Financial implications of New York City's weight management initiative

Sagun Agrawal[1], Alexis C. Wojtanowski[2], Laura Tringali[2], Gary D. Foster[2,3], Eric A. Finkelstein[1] *

1 Program in Health Services & Systems Research, Duke-NUS Medical School, Singapore, Singapore, 2 WW International (formerly Weight Watchers), New York, NY, United States of America, 3 Center for Weight and Eating Disorders, Perelman School of Medicine, University of Pennsylvania, Philadelphia PA, United States of America

* eric.finkelstein@duke-nus.edu.sg

## Abstract

### Purpose

To estimate potential annual savings in medical expenditures from a subsidized weight management program from the NYC Government perspective.

### Design

Longitudinal observational study.

### Setting

Employees of New York City (NYC) government and enrolled dependents.

### Sample

14,946 participants with overweight and obesity.

### Intervention

WW (formerly Weight Watchers®) 'Workshop' and 'Digital' programs.

### Measures

Participation rate, enrollment duration, weight change, and predicted gross and net total and per capita medical expenditure savings and return on investment (ROI).

### Analysis

Participation rate, enrollment duration, weight change, and program costs are based on direct observation. Predicted savings are simulated based on published data relating weight loss to medical expenditure reductions.

**Data Availability Statement:** Due to the contractual requirements by the City of New York we are unable to share this data without a data use agreement in place. If a requester wishes to

replicate this work, they can send a data request to Ratna Singh (ratna.singh@duke-nus.edu.sg) or Eric Finkelstein (eric.finkelstein@duke-nus.edu.sg).

**Funding:** As noted in the original submission, this work was funded by WW International, a global wellness company. We now also note the following: All authors, including two from the funder, played a role in the conception and design of this study. Agrawal conducted all data analyses with input and oversight from Finkelstein. These analyses were conducted independent of the funder. Finkelstein and Agrawal led manuscript writing with input from all authors. Publication of results was agreed at the time of execution of the contract and independent of what the final results showed.

**Competing interests:** Authors Wojtanowski, Tringali, and Foster are employees of WW International. Finkelstein is a paid consultant and paid member of the WW International Scientific Advisory Board. These competing interests do not alter our adherence to PLOS ONE policies on sharing data and materials.

## Results

In total, 47% of participating employees and 50% of participating dependents lost weight during the enrollment period. Mean (median) enrollment duration for employees was 7.1 months (7.0) and for dependents was 6.9 months (6.0). Mean (median) weight losses for the employees in 'Workshops' and 'Digital' was 6.6 lbs (2.80) and 6.3 lbs (0.0). For dependents, weight losses were 7.4 lbs (3.59) and 11.6 lbs (2.0). Per capita and total predicted net savings to NYC Government from employees was estimated to be $120 and $1,486,102 for an ROI of 143%. Including dependents, predicted net savings increases to $1,963,431 for an ROI of 189%. Over 80% of savings came from participants in the Obese III category.

## Conclusion

An evidence-based weight management program has the potential to generate a positive ROI for employers. Future studies should validate these estimates using actual data and more rigorous designs.

## Introduction

Obesity has reached epidemic proportions globally [1]. Excess weight is associated with increased risks for diabetes, coronary heart disease, certain forms of cancer, and other health conditions [2]. As a result, medical expenditures incurred to treat obesity-related conditions are significant, accounting for 9.1% of total annual U.S. medical expenditures [3].

Studies have shown that, among those with excess weight, weight loss of 2.5% or more can generate clinically meaningful health improvements [4–6] and prevent short term deterioration in Health-Related Quality of Life (HRQOL), with greater weight loss leading to greater improvements [7]. Cawley et al. [8] used nationally representative data and econometric analyses to estimate medical costs savings resulting from weight loss of 5% or greater for those with starting BMI values ranging between 30 kg/m$^2$ and 45 kg/m$^2$. They report that, even with just 5% weight loss, the estimated annual savings in medical costs are $US69 in 2010 dollars for those with a starting BMI of 30 kg/m$^2$, $US528 for those with a starting BMI of 35 kg/m$^2$, and $US2,137 for those with a starting BMI of 40 kg/m$^2$. In most cases savings are predicted to increase with greater weight loss. Therefore, depending on the costs to achieve the weight loss, successful weight loss programs have the potential to show a positive return on investment (ROI), even when limited to savings in medical expenditures.

Recognizing the increased risk of chronic diseases and financial implications of excess weight, the City of New York Health Benefits Program and the unions, as represented by the Municipal Labor Committee (MLC), embarked on an effort to bring a weight management program to employees and dependents. They chose to offer a 50% subsidy on membership for WW (formerly Weight Watchers®) to eligible employees. Dependents could also join the program but without any subsidy. WW provides a weight management program available in person and/or digitally. The WW program has been shown both to produce clinically meaningful weight loss [9–15] and be cost effective [16].

The goal of this study is to provide an estimate of the per capita and total annual savings in medical expenditures and return on investment, from the City of New York (NYC) perspective, that may result from subsidized access to WW for employees and non-subsidized access for dependents. We report the percentage of employees who participated in the program,

mean and median weight change among participating employees and dependents, and an estimate of predicted savings by applying the weight change and enrollment duration to the savings estimates presented in Cawley et al. [8]. We then compare these savings to NYC program costs for subsidized access for employees to report the net annual medical expenditure savings and ROI, assuming that all weight change is attributable to the subsidy.

## Materials and methods

### Design

This is a longitudinal observational study where participants' weight data were collected between June 1, 2016 and August 31, 2017. The perspective is the New York City Government. The study procedure is in accordance with the Declaration of Helsinki and was approved by the National University of Singapore Institutional Review Board Reference Code: S-18-216E. Written consent was obtained from all participants.

### Sample

The sample consisted of employees and dependents of NYC. The total eligible employee population is approximately 430,000 individuals. Individuals who are employed by one of the 150 collective bargaining units and eligible for health benefits under the NYC Health Benefits Program and with a BMI of 26 kg/m$^2$ or more were eligible for WW program enrollment. The type of employees eligible includes administrative staff, healthcare providers, teachers, emergency services personnel, sanitation services, among others. Dependents included spouses, domestic partners, and other adult dependents.

### Intervention

Employees had a choice to enroll into WW Workshop+ Digital (termed Workshops) or WW Digital [17] through a NYC portal at subsidized prices. Dependents also had the option to enroll for WW through the NYC portal but at no subsidy. The enrollment was on a rolling basis, therefore each participant had between 1 and 12 months of potential access to the intervention during the analysis period.

Participants who signed up for Workshops were provided with access to weekly 30–45 minute in-person sessions facilitated by trained WW coaches at their workplace or in the community at a location and time convenient to them. At the start of each session, participants had a private check-in with a WW coach and were given the opportunity to weigh themselves, reflect on their progress, and set actionable steps for the next week. Then the weekly group discussion began with celebrating and problem-solving member's successes and challenges over the past week. After that, a new topic and technique focused on a skill related to weight loss and behavior change was introduced. Workshop participants also had access to a suite of digital tools that include self-monitoring of intake, activity, and weight, informational articles, virtual support from WW coaches through one-on-one online chats available 24 hours a day, and an online social media peer community limited to WW members. Participants who signed up for Digital were provided with access to the same suite of digital tools as Workshop participants but received all intervention materials electronically.

### Measures

Measures included: (i) Participation rate (enrolled/eligible) of employees and count of employees and dependents who enrolled in WW Workshop and Digital programs; (ii) Duration of enrollment. This is the period that costs are accrued for employees; (iii) Participation duration

for employees and dependents. This is the period that weights are actively being reported and thus the period that participants are eligible to generate savings. This period may be less than the duration of enrollment as some participants continue to pay but do not actively participate and/or report weights; (iv) Weight change for employees and dependents enrolled in the workshops and Digital programs respectively; (v) Predicted savings in annual medical expenditures due to weight change for employees and dependents; (vi) WW Program costs for employees; and (vii) Predicted annual net savings for NYC and predicted ROI.

## Data

Participant-level data from the WW database (self-reported date of birth and height at enrollment), measured (for Workshop participants) or self-reported (for Digital participants) weights, program enrollment and cancellation dates, NYC costs for the WW program, and estimated annual medical expenditure savings associated with weight change from Cawley et al. were used to conduct the analysis.

## Analysis

Participation rate was calculated as the percentage of employees offered the subsidy who signed up for one of the two programs. We do not report a rate for dependents as we did not have access to the total number of dependents eligible for the programs. Duration of enrollment was calculated as the duration between the enrollment date and the disenrollment date or the end of the evaluation period (May 31, 2017), whichever came first. If a participant did not have a disenrollment date, she is assumed to be enrolled and incurs costs till the end of the evaluation period. This assumption is reasonable as unless an employee actively dis-enrolls, she continues to pay 50% of the costs of participation, regardless of whether she actively participates.

Participation duration is a subset of duration of enrollment. This tracks the period that the participant is reporting weight outcomes, and thus has the potential to generate savings. It was calculated as the duration between the enrollment date and the date of the last weigh-in during the evaluation period (June 1, 2016 to May 31, 2017), with one exception. If a participant has a weigh-in during June 2017 to August 2017 she is assumed to still be actively participating and the last date of her enrollment during the year is set to May 31, the last day of the evaluation period. If a participant had less than two weights recorded, she was assigned a participation duration of zero. Therefore, participation duration ranges between 0 and 12 months for all participants.

For those with two or more weights recorded during the evaluation period, weight change was calculated as the difference between the last and first recorded weight, with one exception. If the participant has an additional weight in the three months following the end of the evaluation period, the last weight is calculated as a weighted average of the two weights closest to (before and after) the end of the evaluation period. This allows for capturing weight changes that are likely to be occurring through the end of the period. An example of this approach is shown in the supporting information S1 File.

Predicted savings were estimated at the participant level by applying each participant's starting BMI, weight loss and participation duration to the savings estimates presented in Cawley et al., after updating them from 2010 to 2017 dollars using the CPI-Medical healthcare inflation index [18]. Cawley et al. reports annual savings in medical expenditures for adults aged 24 to 65 for BMI reductions of 5, 10, 15, and 20% based on starting BMIs between 30 and 45 kg/m$^2$. The model aggregated medical expenditures over all types of medical care including inpatient and outpatient medical care, emergency visits and prescription drugs. Unlike most

obesity modelling studies, their modelling strategy includes an instrumental variables approach designed to address measurement error and omitted variable bias, and thus was deemed to be the best source of data for this study.

We make several assumptions to allow the Cawley estimates to be applied to the weight change data among NYC WW participants. These assumptions are as follows:

1. Cawley et al. only reported savings for those with weight loss of 5% or greater. We linearly extrapolated savings for all those with BMI reduction (i.e., weight loss) between 2.5% and 5%. For example, if an individual had a 2.5% reduction in BMI then she would receive medical cost savings of half of the corresponding value in the 5% BMI reduction column in Cawley et al. Table 1. Clinical evidence suggests that weight loss as little as 2.5% generates clinical health improvements [4–6] thus, we deemed it reasonable to apply cost savings for weight losses of this magnitude or greater, although we explore the implications of this assumption in the sensitivity analysis. For weight losses smaller than 2.5% no savings were assumed.

2. For each starting BMI, Cawley et al. present savings for weight losses of 5%, 10%, 15% and 20%. For weight losses between these values, we estimated savings using linear interpolation.

3. Cawley et al. did not provide savings estimates for those in the overweight range (BMI values between 25 and 29.9 kg/m$^2$) or for those with BMI values above 45 kg/m$^2$. We assumed $0 savings for those with a starting BMI in the overweight range due to lack of evidence that weight loss translates into reductions in medical expenditures for this BMI category. For the 5.7% of participants with starting BMI values above 45 kg/m$^2$ *(n = 858)* we conservatively recoded their BMI to 45 kg/m$^2$ and assigned savings as reported in Cawley et al. based on the amount of weight lost.

4. Cawley et al. also do not report savings for those with greater than 20% weight loss. Therefore, for those few cases with weight change greater than 20% *(n = 58 for weight loss, n = 7 for weight gain)* we recoded the change to 20%.

5. Cawley et al. present annual savings. We adjust this amount based on the participation duration (i.e., 6 months of participation duration accrues only 50% of annual savings) but assume that no savings accrue for participation duration less than 3 months. Evidence suggests that health improvements from diet, exercise, and weight loss can begin within weeks [19], therefore minimum participation duration of three months to accrue savings is a reasonable assumption. However, in the sensitivity analyses we also present results assuming that the minimum duration required to generate savings is 6 months.

6. To minimize risks of bias, we apply the estimates in Cawley et al. Table 1 in reverse for those who gained weight. For example, someone with a starting BMI of 33 kg/m$^2$ who lost

**Table 1. Demographic characteristics of sample.**

| Demographics | Employees Mean (S.D.) | Dependents Mean (S.D.) | Total Mean (S.D.) |
|---|---|---|---|
| Age (years) | 44.46 (10.53) | 46.82 (13.12) | 44.85 (11.04) |
| Initial BMI, kg/m$^2$ | 33.93 (5.65) | 33.91 (5.41) | 33.92 (5.61) |
| % Female | 91% | 78% | 89% |
| Membership type | | | |
| % Workshops | 57% | 67% | 58% |
| % Digital | 43% | 33% | 42% |
| No. included in Analysis | 12,436 | 2,510 | 14,946 |

5% of his baseline weight would have a predicted savings in annual medical expenditures of $288 (in 2010 dollars). If a different individual with the same starting BMI gained 5% of their baseline weight we apply the $288 as a cost, not as a savings. This conservative approach addresses concerns due to omitted variables, mean reversion [20], and other potential sources of bias.

7. All predicted savings are assumed to accrue immediately and terminate after the last weigh in. In other words, we do not allow time for savings to accrue nor do we allow time for residual benefits.

8. Because the City of New York's base health plans cover 74.7% of medical expenditures referenced in the Cawley et al. model, we multiply each participant's predicted savings by 0.747 to reflect savings from the NYC government perspective.

We present predicted savings stratified by starting BMI, BMI reduction, and program type separately for both employees and dependents.

WW program costs from the NYC perspective were calculated based on membership rates of $30 per month for Workshops and $14 per month for Digital. NYC paid 50% of these costs for employees but did not pay for dependents. After applying the 50% subsidy for employee costs and assuming no costs for dependents, we present per capita gross and net savings and the ROI %, defined as net savings divided by NYC costs, associated with subsidized access to WW for NYC employees and non-subsidized access for dependents. A more detailed description of the methods, including several example calculations, is available in the supporting information S1 File.

In efforts to gauge the sensitivity of the ROI results to key model assumptions, and given the inability to access the raw data presented in Cawley et al. for producing confidence intervals around the estimates, we present a series of one-way sensitivity analyses. The key assumptions are 1) minimum weight loss to generate medical cost savings, 2) minimum duration that weight loss must be maintained to generate savings, and 3) the magnitude of savings for any given starting BMI/weight loss. To gauge the influence of results on these key assumptions, we present alternative ROI estimates under the following more conservative scenarios:

1. Minimum weight loss to generate expenditure savings is 5% as opposed to 2.5%,

2. Minimum duration to generate expenditure savings is 6 months as opposed to 3 months, and

3. Medical expenditure savings are 50% of what is reported in Cawley et al. Table 1

## Results

### Participation

19,371 participants, including 4.5% of eligible employees, enrolled into the program. However, due to exclusions (Fig 1), only 12,436 employees and 2,510 dependents were eligible for the analysis. Among eligible employees, 57% signed up for 'Workshop' and the remainder (43%) signed up for 'Digital'. For dependents, these percentages were 67% and 33%.

### Sample characteristics

Table 1 shows the demographic characteristics of the sample. The average age was 44.5 and 46.8 years for employees and dependents, respectively, and the average starting BMIs were 33.9 kg/m$^2$ for both; 91% of employees and 78% dependents in the sample were female.

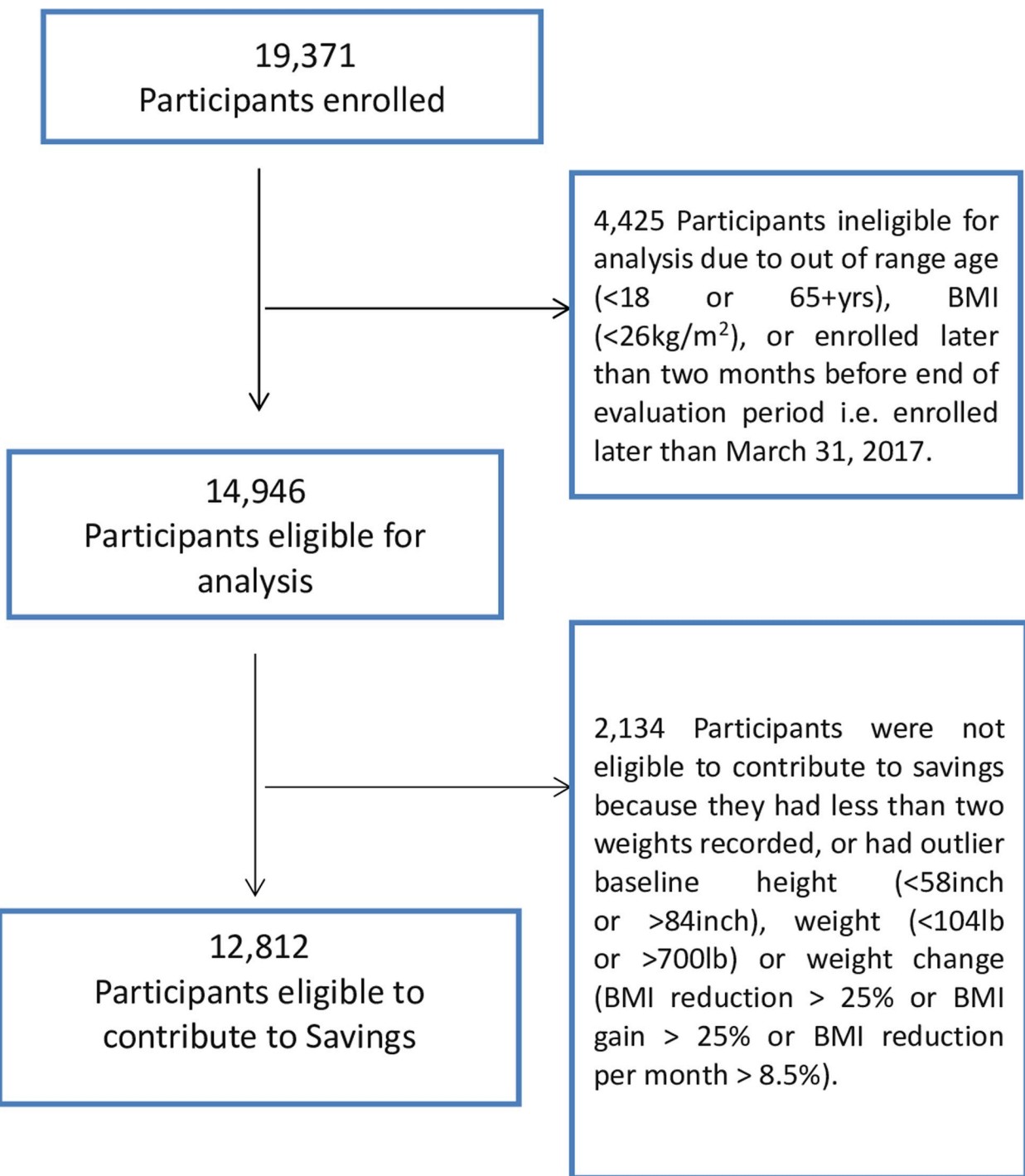

**Fig 1. STROBE diagram.**

### Duration

The average enrollment duration for employees was 7.1 months (median 7.0) and for dependents was 6.9 months (median 6.0). However, the participation duration, the average time between first and last weigh-in was much shorter; 3.7 months (median 2.27) for employees

and 3.9 months (median 2.60) for dependents; 44% of employees and 38% of dependents did not report two weights more than three months apart, thus they were not eligible to generate savings. Note that since enrollment was rolling not everyone in the sample had a chance to be enrolled for the entire 12 months and those enrolled in the final three months generated costs but were unable to generate savings by construction.

## Weight change

During the evaluation period, the mean weight loss for employees enrolled in 'Workshops' was 6.6 lbs (median 2.80) and for 'Digital' was 6.3 lbs (median 0.0), unadjusted for enrollment duration. For dependents, the mean weight loss for 'Workshops' was 7.4 lbs (median 3.59) and for 'Digital' was 11.6 lbs (median 2.0). Fig 2 shows that, among employees, 11% had 2.5 to 5% weight loss, 10% had 5 to 7.5% weight loss, 4% had 7.5 to 10% weight loss and 8% had more than 10% weight loss. For dependents, 12% had 2.5 to 5% weight loss, 9% had 5 to 7.5% weight loss, 4% had 7.5 to 10% weight loss and 9% had more than 10% weight loss. In total, 46.7% of participating employees and 50.4% of participating dependents lost weight during the enrollment period.

## Predicted medical expenditure savings

Over the 12-month analysis period, predicted mean per capita medical expenditure savings to NYC for employees was $276 (median 0.0) and $108 (median 0.0) for 'Workshops' and 'Digital' respectively. The corresponding mean per capita savings for dependents was $238 (median 0.0) and $93 (median 0.0). Total predicted savings for employees was $1,940,050 and $582,479 for 'Workshops' and 'Digital' respectively. For dependents, these figures were $400,115 and $77,214. In aggregate, gross savings to NYC was predicted to be $2,999,858. Of this total, 84% came from employees, with dependents contributing 16%.

By program type (Table 2), among those employees eligible to contribute to savings (N = 10,766) 77% of gross predicted savings was attributable to the 55% of participants in 'Workshops' and the remaining 23% was attributable to the 45% in 'Digital'. For dependents who were eligible to contribute to savings (N = 2,046), 84% was attributable to the 66% of participants in 'Workshops' and 16% was attributable to the 34% in 'Digital'.

By enrollment BMI category (Table 3), for employees who were eligible to generate savings, 84% of savings came from the 18% of participants who belong to the Obese III group; 13% came from the 22% in the Obese II group; 3% came from the 33% in the Obese I group. By construction, overweight participants were assumed not to generate savings. The distribution for dependents is similar.

By BMI reduction (Table 4), for employees who were eligible to generate savings, 56% came from the 8% of participants who lost greater than 10% of their baseline weight; 14% came from the 4% who lost 7.5–10%; 28% came from the 11% who lost 5–7.5% weight, and 19% come from the 12% who lost 2.5–5% weight. Negative savings (i.e., costs) of 17% came from the 15% who gained weight. The distribution for dependents is again similar. Note that Table 4 consists of those eligible to generate savings. However, even if weight loss exceeds 2.5%, eligible participants may not generate savings. This would occur if they do not meet other criteria, including participation duration less than 90 days and/or enrollment BMI less than 30 kg/m$^2$. Although 35% of participants had weight losses of 2.5% or more, only 2,426 participants (19%) out of those eligible to contribute to savings actually did so.

## Program costs

Based on each of the 12,436 participating employee's duration of enrollment and the monthly subsidy of $15 for Workshops and $7 for Digital programs, the average per capita and total

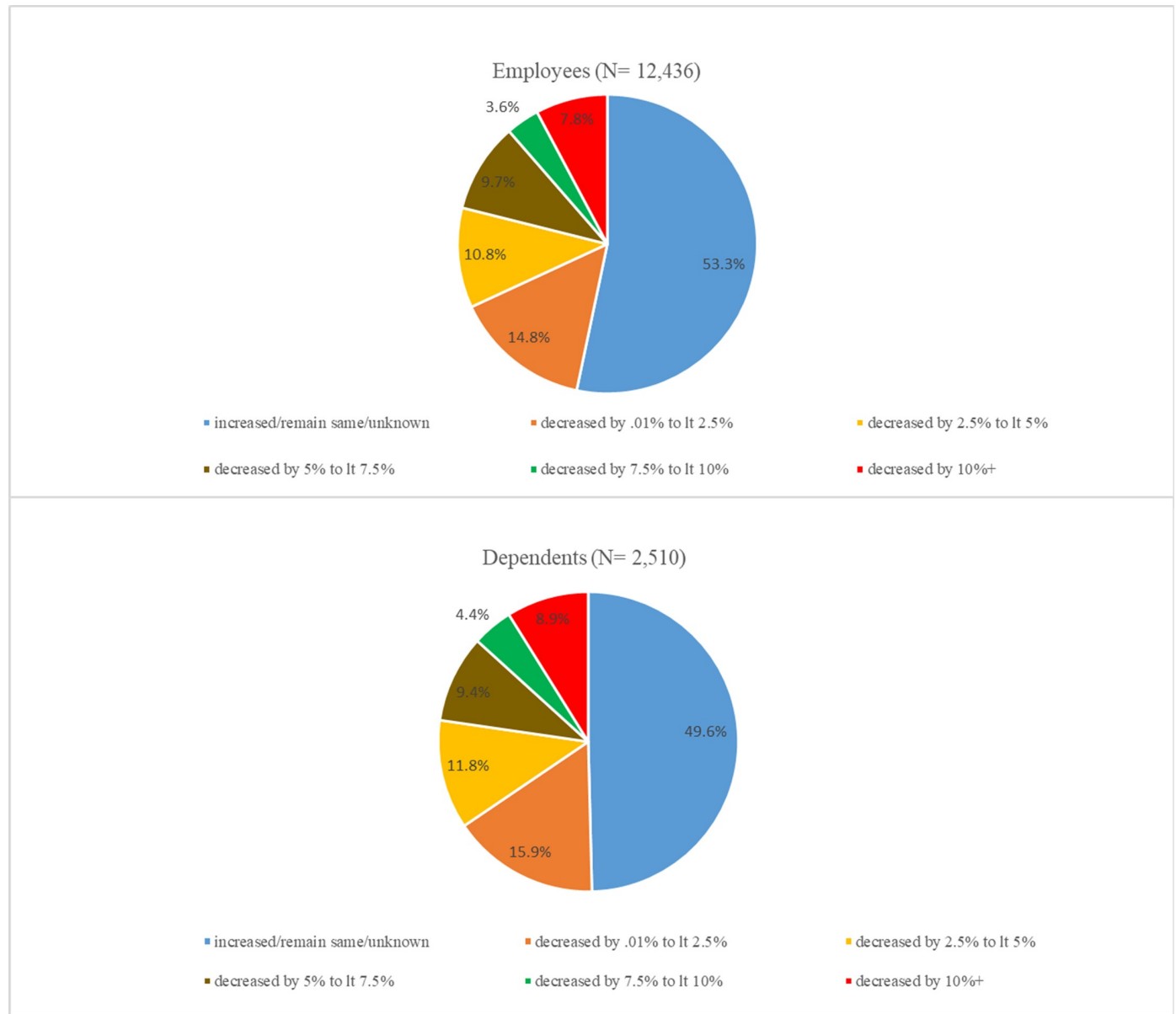

**Fig 2. Distribution of weight change among participants.**

costs to NYC to subsidize the WW program for employees was $111 and $47 and $781,830 and $254,600 for Workshops and Digital respectively (Table 5).

## Return on investment

Net savings are shown in supporting information S1 Table. After subtracting WW program costs from predicted gross savings, the mean per capita and total net savings for employees was $120 and $1,486,102, resulting in a ROI of 143%. This ROI is equivalent to an investment of $100 generating revenue of $143 one year later. Including dependents, who incur no costs to the city, total net savings increases to $1,963,431 for a ROI of 189%.

**Table 2. Gross savings by program type.**

| Gross Savings by Program Type | Digital | Workshops | Total |
|---|---|---|---|
| *Employees (N = 10,766)** | | | |
| Gross Savings ($) | $582,479 | $1,940,050 | $2,522,529 |
| % of Total Participants | 45% | 55% | 100% |
| % of Gross Savings | 23% | 77% | 100% |
| *Dependents (N = 2,046)** | | | |
| Gross Savings ($) | $77,214 | $400,115 | $477,329 |
| % of Total Participants | 34% | 66% | 100% |
| % of Gross Savings | 16% | 84% | 100% |

* This sample consists of those eligible to generate savings.

## Sensitivity analysis

The base case analysis assumed minimum weight loss to generate expenditure savings is to 2.5%, minimum duration to generate expenditure savings is 3 months, and savings are as they appear in Cawley et al. Table 1. Table 6 below presents results using the more conservative assumptions for key input parameters.

The difference in total gross savings between the Base Case and Case 2 of roughly $316,000 results from the removal of savings for 12.3% of participants who lost between 2.5% and 5% of baseline weight and the removal of costs (i.e., negative savings) for 3.8% who gained weight in this range. The difference of roughly $634,000 between the Base Case and Case 3 results from the removal of savings (negative savings) for the 18% (1.1%) of participants whose enrollment duration was between 3 and 6 months. The final column of Table 6 reveals that even after assuming costs are 50% of what they appear in Cawley et al., per capita net savings and ROI remain positive at $31 and 44% respectively.

## Discussion

### Summary

The City of New York Health Benefits Program and the unions, as represented by the MLC, offered a 50% subsidy to eligible employees to join a commercially available weight management program, WW. We estimate that, even when limiting savings to the 76% of employees with enrollment BMI values 30 kg/m$^2$ and above, the forecast ROI associated with the subsidy was 143%. This ROI was achieved even though only 3% of employees enrolled in the program

**Table 3. Gross savings by enrollment BMI.**

| Gross Savings by Enrollment BMI | Overweight (BMI 26.0–29.9) | Obese I (BMI 30.0–34.9) | Obese II (BMI 35.0–39.9) | Obese III (BMI ≥ 40) | Total |
|---|---|---|---|---|---|
| *Employees (N = 10,766)** | | | | | |
| Gross Savings ($) | $0 | $78,584 | $328,631 | $2,115,314 | $2,522,529 |
| % of Total Participants | 27% | 33% | 22% | 18% | 100% |
| % of Gross Savings | 0% | 3% | 13% | 84% | 100% |
| *Dependents (N = 2,046)** | | | | | |
| Gross Savings ($) | $0 | $18,422 | $75,558 | $383,348 | $477,329 |
| % of Total Participants | 24% | 36% | 23% | 17% | 100% |
| % of Gross Savings | 0% | 4% | 16% | 80% | 100% |

* This sample consists of those eligible to generate savings.

Table 4. Gross savings by BMI reduction.

| Gross Savings by BMI Change | weight gain | no change in weight | .01% to <2.5% Weight Loss | 2.5% to <5% Weight Loss | 5% to <7.5% Weight Loss | 7.5% to <10% Weight Loss | 10%+Weight Loss | Total |
|---|---|---|---|---|---|---|---|---|
| *Employees (N = 10,766)** | | | | | | | | |
| Gross Savings ($) | $-430,286 | $0 | $0 | $469,226 | $716,772 | $361,495 | $1,405,321 | $2,522,529 |
| % of Total Participants | 15% | 34% | 17% | 12% | 11% | 4% | 8% | 100% |
| % of Gross Savings | -17% | 0% | 0% | 19% | 28% | 14% | 56% | 100% |
| *Dependents (N = 2,046)** | | | | | | | | |
| Gross Savings ($) | $-70,874 | $0 | $0 | $66,749 | $150,106 | $64,228 | $267,120 | $477,329 |
| % of Total Participants | 17% | 25% | 19% | 14% | 11% | 5% | 9% | 100% |
| % of Gross Savings | -15% | 0% | 0% | 14% | 32% | 13% | 56% | 100% |

* This sample consists of those eligible to generate savings.

and only 1/3rd of those lost enough weight and over a long enough duration to generate savings. When examining employees and dependents combined, the ROI increased to 189%.

The low program uptake reflects the reality that most employees are not necessarily interested in signing up for a behavioral weight management program and/or that efforts to make the program known were only partially successful. The fact that only 47% of enrollees lost weight may be because some found the program not to be a good fit for them and/or because the challenges of their personal lives and professional roles made it difficult to maintain the changes in behavior necessary for weight loss. Regardless, results reveal that, due to the high costs of obesity, even moderate weight loss among a subset of enrollees would be enough to generate a positive return on investment. This results because, among the 19% who generated savings, savings were large. For these individuals, average program participation duration was 7.24 (median 7.02) out of a possible 12 months. Moreover, although program costs for these individuals averaged $90, due to average weight losses of 17.5lbs, medical expenditure savings averaged $1,422. This was more than enough to offset the costs for those whose starting BMI was too low to generate savings (i.e. overweight range), were unsuccessful in their weight loss efforts and/or who joined too late in the year to generate savings. However, it is important to point out that over 80% of the predicted savings came from participants in the Obese III category. This result is not surprising given that this group is most expensive upon enrollment and, therefore, has the most to benefit from successful weight loss. However, it suggests that the distribution of those who join the program will greatly influence the ROI potential. Although this may suggest that the city (or others) should not invest in those in lower BMI categories, including overweight individuals who we assume generate no savings, this is not necessarily the case. Given that individuals tend to gain weight as they age [21], a dynamic

Table 5. Program costs.

| Program Costs | Digital | Workshops | Total |
|---|---|---|---|
| *Employees (N = 12,436)* | | | |
| Total Costs | $254,597 | $781,830 | $1,036,427 |
| % of Total Participants | 44% | 56% | 100% |
| % of Total Costs | 25% | 75% | 100% |
| *Dependents (N = 2,510)* | | | |
| % of Total Participants | 33% | 67% | 100% |

**Table 6. Sensitivity analysis.**

| | Base Case | 5% vs. 2.5% min weight loss (2) | 6 months vs. 3 months min duration (3) | Savings at 50% of Base Case (4) |
|---|---|---|---|---|
| Total Gross Savings | $2,999,858 | $2,683,943 | $2,366,293 | $1,499,929 |
| Per Capita Savings | $201 | $180 | $158 | $100 |
| Total Costs | $1,036,427 | $1,036,427 | $1,036,427 | $1,036,427 |
| Per Capita Costs | $69 | $69 | $69 | $69 |
| Total Net Savings | $1,963,431 | $1,647,516 | $1,329,866 | $463,502 |
| Per Capita Net Savings | $131 | $110 | $89 | $31 |
| ROI | 189% | 159% | 128% | 44% |

analysis may show that early interventions, such as WW, may not generate savings in the short-term but are effective at avoiding future costs and thus may show a positive ROI when cost avoidance is considered. This should be an area of future research.

## Strengths and limitations

This study has many strengths, including reliance on real world weight loss data to simulate the ROI from an evidence-based weight program among a large and heterogeneous sample of employees and dependents. However, the analysis is also subject to several limitations. A primary limitation is that the savings are simulated based on an existing model and not based on actual data. This is necessary given a lack of data at the participant level that ties weight loss to medical expenditures. For simplicity, we further assumed that weight loss translates immediately to savings if maintained for three months or longer but ceases immediately after the last weigh in. In reality, it may take time for the savings to accrue and there are likely residual benefits beyond the last measurement, perhaps even if the weight is regained. Our changes in medical expenditure are limited to those with BMI values above 30 kg/m$^2$ at baseline due to lack of evidence that weight loss translates into reductions in medical expenditures for overweight individuals. For the remaining sample, our analysis assumes that all weight loss results in reductions in medical expenditures. However, there is evidence that if the weight loss is large enough, such that BMI values fall below 25 kg/m$^2$, medical expenditures may increase. However, for those with starting BMI values above 30 kg/m$^2$ in our data, only 0.05% lost enough weight such that their final BMI was below 25 kg/m$^2$ at follow up.

Although we conservatively applied costs for those who gained weight, we assumed that BMI increases of a given percentage have the same effect on annual medical expenditures, in absolute terms, as BMI reductions. In reality, risks of chronic disease and medical expenditures increase non-linearly with increasing BMI. Therefore, our cost increases for those who gained weight may be underestimated. This is further exacerbated by applying zero costs for overweight individuals who gained weight. However, given only 15% of our sample, including overweight participants at baseline, gained weight and average weight gain was only 5.7lbs, varying these assumptions is likely to have little impact on the resulting ROI estimate. Other limitations are that only self-reported weights were available for Digital participants. The sample is also largely female and there are no data on race or ethnicity, thus limiting our ability to conduct analyses on subgroups of interest.

Importantly, although these results represent the best estimates of the costs, savings, and ROI associated with the subsidized program, we cannot say the program caused the savings. This results, because, in the absence of the subsidy, some employees would have made efforts to lose weight either by joining WW and paying on their own, or in myriad other ways, and some would have been successful. Therefore, these results may represent an upper bound of

the potential savings resulting from the subsidized weight management program. These estimates are also associated with great uncertainty. In efforts to gauge the influence of key parameters on the estimates, we present savings and ROI estimates both by applying stratifications that allow for determining which population subsets are most influential in the ROI calculation and by including one-way sensitivity analyses of key input parameters. Even in our most conservative scenario of savings reduced by 50%, the forecast ROI remained positive. Future studies could improve on these estimates by using linked data on weight change and changes in medical expenditures using more rigorous randomized trial designs that are free from selection and the other potential biases and that can incorporate both parametric and non-parametric sensitivity analyses. These studies should also consider the effects of weight loss on work productivity, as both absenteeism and presenteesim (reduced productivity while working) are greater among those with excess weight and are expected to diminish with successful weight loss [22,23].

## Supporting information

**S1 Table. Net savings.**
(DOCX)

**S1 File. Steps of analysis.**
(DOCX)

**S2 File. STATA code.**
(RTF)

## Author Contributions

**Conceptualization:** Eric A. Finkelstein.

**Data curation:** Alexis C. Wojtanowski, Laura Tringali, Gary D. Foster.

**Formal analysis:** Sagun Agrawal, Eric A. Finkelstein.

**Methodology:** Eric A. Finkelstein.

**Project administration:** Alexis C. Wojtanowski, Laura Tringali, Gary D. Foster.

**Writing – original draft:** Sagun Agrawal, Eric A. Finkelstein.

**Writing – review & editing:** Sagun Agrawal, Alexis C. Wojtanowski, Laura Tringali, Gary D. Foster, Eric A. Finkelstein.

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
