## [Decision Letter · Decision Letter 0]

3 Jan 2020

PONE-D-19-29326

Financial Implications of New York City's Weight Management Initiative

PLOS ONE

Dear Dr. Finkelstein,

Thank you for submitting your manuscript to PLOS ONE. After careful consideration, we feel that it has merit but does not fully meet PLOS ONE’s publication criteria as it currently stands. Therefore, we invite you to submit a revised version of the manuscript that addresses the points raised during the review process.

The reviewers have noted concerns with the assumptions, the statistical reporting and the conclusions derived from these.  The editor suggests that the results may not be extraordinary if the assumptions are revisited, however, even if they are not, reporting the results are interesting.  Toning down the claims of the program as suggested by the reviewers is critical.

We would appreciate receiving your revised manuscript by Feb 14 2020 11:59PM. To enhance the reproducibility of your results, we recommend that if applicable you deposit your laboratory protocols in protocols.io, where a protocol can be assigned its own identifier (DOI) such that it can be cited independently in the future. For instructions see: http://journals.plos.org/plosone/s/submission-guidelines#loc-laboratory-protocols

We look forward to receiving your revised manuscript.

Kind regards,

Diana M. Thomas

Academic Editor

PLOS ONE

Journal Requirements:

4. Thank you for stating the following in the Financial Disclosure section:

This work was supported by WW International.

We note that one or more of the authors are employed by a commercial company: WW International.

Additional Editor Comments:

The reviewers have noted concerns with the assumptions, the statistical reporting and the conclusions derived from these. The editor suggests that the results may not be extraordinary if the assumptions are revisited, however, even if they are not, reporting the results are interesting. Toning down the claims of the program as suggested by the reviewers is critical.

Reviewers' comments:

Reviewer's Responses to Questions

**Comments to the Author**

1. Is the manuscript technically sound, and do the data support the conclusions?

Reviewer #1: Yes

Reviewer #2: No

Reviewer #3: No

Reviewer #4: Partly

2. Has the statistical analysis been performed appropriately and rigorously? 

Reviewer #1: Yes

Reviewer #2: No

Reviewer #3: No

Reviewer #4: No

3. Have the authors made all data underlying the findings in their manuscript fully available?

Reviewer #1: No

Reviewer #2: No

Reviewer #3: No

Reviewer #4: Yes

4. Is the manuscript presented in an intelligible fashion and written in standard English?

Reviewer #1: Yes

Reviewer #2: Yes

Reviewer #3: Yes

Reviewer #4: Yes

5. Review Comments to the Author

Reviewer #1: This is a well written paper to quantify the annual medical expenditure savings from an employee subsidized WW workshop or digital program weight management intervention. Overall, I believe this is a well conducted study with limitations noted in the discussion and abstract. Needed revisions are listed below.

Abstract:

1) Important to highlight that only 51% of employee and 54.1% of dependent enrollees were eligible to generate savings

2) If space allows, would also be good to highlight that the majority of savings (73% for both employees and dependents) came from enrollees with class II or III obesity.

Methods:

3) Line 64. Note that height was also collected.

4) Line 65. Awkward sentence regarding “the perspective is NYC”. Reword.

5) Unclear how they accounted in the ROI for employee and dependent duration of enrollment. Did the costs for individuals included differ for those enrolled four months (and employee subsidy covered 4 months) versus those enrolled six months (and employee subsidy covered 6 months)? This was not clear from the methods described.

6) Line 163. What did the authors do for individuals with a BMI between 29.1 and 29.9?

7) Line 183-185. The authors do not provide justification for why in the sensitivity analysis, they estimate savings assuming that weight loss is maintained throughout the duration of the enrollment. Why is that assumption made given the known likelihood of weight regain once the intervention ends?

8) In Figure 1, the authors describe participants that were not eligible because they had an outlier for baseline height, weight, or weight change. Please add a description of what those cutoffs were, the justification for them, and how this was conducted.

Results:

9) Line 247-248. This sentence is confusing.

Discussion:

10) Additional discussion is needed on the fact that only 51% of employee and 54.1% of dependent enrollees were eligible to generate savings and related limitations.

11) Additional discussion is needed on the fact that the majority of savings (73% for both employees and dependents) came from enrollees with class II or III obesity and related implications.

12) There was a great deal of diversity related to employee types including administrative staff, healthcare providers, teachers, emergency services personnel and first responders, sanitation and janitorial services, among others. Do the authors have data related to job type, income, or any other indicator of socioeconomic status and how that may play a role in weight loss efforts of the employees and dependents?

Minor comments:

13) There are a great deal of typos throughout the manuscript that need to be carefully checked. Some examples include:

a. Line 90. Add ‘s’ to the end of behavioral skill.

b. The word enrollment is spelled incorrectly throughout the entire manuscript (it is spelled enrolment).

c. Line 174. Comma at the end of the sentence rather than a period.

d. Line 245. Should be a space between SD and the number.

e. Line 289. Typo after analysis.

14) Line 211. Should be clear that it is 91% of enrolled employee and dependents that were female.

Reviewer #2: ** Main concern **

[Lines 157] "We assigned zero savings to those who gained weight ..." Unlike the last two items in the next section, this definitely has a structural, biasing impact on the results and is definitely unreasonable. If someone started a program and gained 10 BMI units, that's just "no savings" to the state? Not a hefty negative savings?

** Other concerns **

[Data] If "Data cannot be shared publicly because of legal issues. Data are available from WW International for researchers who meet the criteria for access to confidential data.", then the data are not "available without restriction"

[Line 148 and beyond] The emphasis that the Cawley estimates are "causal" (note the deliberate italics in line 148), is questionable:

1) An instrumental variable approach is just a mediational analysis with some extra assumptions; without those assumptions, there's no causality

2) Cawley et al. do spend time addressing/justifying this point (see, for example, the passage that mentions their references 30 and 31), but (a) reference 30 is to an entire book, which is rarely a sign of argumental strength and (b) reference 31 estimates the shared environment effect at 10%, which is not nothing

3) Lastly, to nitpick, Cawley et al. are discussing weight as an IV, not BMI

So while the argument for causality under the Cawley et al. model is stronger than the majority of other such cases where causality is invoked, this still needs some caveating by the authors.

[Lines 166-167] Item #3 in this passage seems arbitrary, unnecessary and could have a big impact on the results. Justify this choice or redo the analyses without it

[Lines 236-238] Similarly, the fact that some dependents were deemed to not generate savings based on having "outlier data" is suspect.

** Lesser issues **

[Line 65] "The perspective is NYC." This statement may make sense in the authors' discipline, but here it is a non sequitur. Please make clear what it is that you are trying to convey with this passage

[Line 82] "Dependents were free to also ..." Considering that this passage is talking about costs, and the costs for dependents is not zero (free), perhaps the authors should use a different phrasing

[Lines 89-90] What is meant by "discussions on behavioral skill"?

[Lines 100-102] It is unclear how (ii) and (iii) differ; the passages later on that describe these don't really clear this up either. Please clarify their distinction

[Lines 159-165] "for those with a BMI below 26" covers both "normal weight" and "overweight"; why is this breakpoint used?

** Errata **

[Line 36 and beyond] Make sure that "et al." is used, as this is short for et alia

[Line 49 and beyond] Inconsistent trademarking of Weight Watchers (compare to line 7)

[Lines 52 and beyond] Inconsistent spacing before reference numerals; see this line for a self-contained pair of examples

[Lines 99-105 and beyond] Inconsistent capitalization of items in lists; compare (i-v) and (vi) here

[Line 101 and beyond] Multiple instances of the typo: "enrolment"

[Lines 108-113] This whole paragraph is a sentence fragment

[Line 108] The parenthetical begun with "(self-reported ..." is never closed

[Line 121] Typo: "date,(s)he" (missing a space)

[Line 160] The period is part of the superscript in "m2."

[Line 174] Typo: "above,."

[Line 202 and beyond] "consort" should be "CONSORT"

Reviewer #3: The manuscript “Financial Implications of New York City's Weight Management Initiative” aimed to fill an important gap about the cost effectiveness in terms of health expenditures of widespread implementation of a commercial weight loss program, WW. Unfortunately, the model includes some fatal assumptions, some highly questionable assumptions, and inadequate handling of error.

The most substantial and fatal assumption is that individuals can only result in cost savings or no differences in costs. Individuals who have gained weight or lost less than 2.5% of weight were set to zero. However, the authors report that participants lost 6.3 to 11.6 lbs on average, but with substantial standard deviations. If weight change is assumed to be normally distributed (which the use of standard deviation would imply), then between 36% and 40% of participants would have gained weight in the sample. Claiming only success of weight losers and ignoring the costs of weight gainers necessarily biases the cost savings estimate in favor of the intervention. I note that the authors also report that 25.7% (line 232) to 27% (line 236) of participants ‘did not lose enough weight’, which indicates that the distributions are not normally distributed, and thus should not be represented by mean and SD, but still supports that a substantial proportion of the participants might have incurred cost increases rather than savings.

There is also no control group, and workplace wellness program evaluations frequently note selection effects. For instance, the Illinois Workplace Wellness Study. What would these same people have experienced without WW?

Some questionable assumptions are based on the use of the estimates from Cawley et al. Cawley et al. employed an instrumental variable approach to estimating the causal effect of weight loss on medical expenditure savings. The use of the Cawley estimates have limitations (Do they match the present sample? How does their approach to addressing self-report affect the measured vs self-report differences used in the present manuscript? Is the IV even appropriate?), but are not an altogether unreasonable basis to estimate changes in medical costs. However, the present manuscript makes several questionable choices. First, rather than staying true to the original estimates, they interpolate values not reported by Cawley et al., including a 2.5% weight loss threshold that is lower than the 5% reported (it appears there is also a 7.5% threshold from tables that was not reported in the methods), and extrapolating below the BMI of 30 to include a greater n, despite Cawley et al. discussing the non-linearity of expenditures over BMI. Indeed, in Table 1 of Cawley, some of the changes in expenditure are not monotonic across % weight loss within some BMI categories. Second, the present authors chose to impute diabetes status because it was not included in the data available. This approach used a weak imputation, involving only estimating proportions based on probabilities of diabetes based on weight status alone (S1). This does not adjust for other key variables (e.g., the present population is predominantly women), and the performance of such an approach for expenditure modeling is not justified within the paper. More importantly, Table 1 of Cawley includes estimates that are averaged across all adults regardless of BMI status. The big concern is that Table 1 appears to be less favorable to support cost savings compared to Tables 2 and 3. Switching from Cawley’s model using all people to stratifying by diabetes status (without: Table 2; with: Table 3) means that cost INCREASES seen for individuals with a BMI of 30 who lose 20% of weight ($235 cost/yr) are instead estimated as SAVINGS for those without diabetes ($905 savings/yr) and with diabetes ($1865 savings/yr). This raises two concerns: why did the authors chose to use a weak imputation approach to estimate diabetes status when a combined model that did not need diabetes status was included in Cawley et al? And, if the models are that instable just from stratifying by diabetes status, are the estimates reliable for estimating cost savings in turn?

Some procedures were not clarified: is someone who lost 4% rounded to the 5% group, rounded down to the 2.5% group, or interpolated?

It appears that the error in Cawley estimates were not propagated forward into this model. That is, Person A in S1 is estimated to have experienced cost savings of $495.66, but what are the error bounds around that given the substantial error term in the Cawley tables? No estimate of uncertainty was provided for any of the model estimates, other than SD that appears based only on the point estimates from Cawley’s tables. The total estimated costs should also have an error term, and the SD reported for per capita values is only for the SD of mean estimates for each individual, rather than the variability compounded with the error presented in Cawley.

Other error terms are confusing. For instance, given the (fatal) imposition of limiting participants to cost-savings only, the estimate of $83 per capita with an SD of 385.5 implies that 41% of people saved nothing or cost-increased (again, because SD is typically used when assuming symmetry). However, the greatest loss would be $15/mo x 12 mo = $180/yr loss. Consider presenting the data in a different format (median/IQR), or clarify how losses could be greater than $180/yr.

The use of two months as a minimum length between weighings was not justified in the paper.

The inclusion of weights after the end of the evaluation was also not justified. If a weight was collected during the evaluation period, it makes no immediate sense to grab weights outside of the evaluation period.

Although I am highly critical of the assumptions above, I also want to highlight that the authors have included important adjustments: adjusting for inflation; including all costs for participants regardless of estimating cost savings (that is, if people were enrolled but did not contribute weights to be able to estimate costs); limiting cost savings to the months that experienced the weight loss (though this raises the question of whether two months would result in cost savings); limiting analysis (and participation) to those with overweight or greater BMI; adjusting costs by the % of expenditures covered by NYC benefits; and others. S1 was exceedingly helpful to walk through the complexities of the estimation procedures.

Other concerns:

The data availability statement does not match PLOS guidance. The answer should be ‘no’ followed by an explanation in the ‘describe where the data may be found…’ field. There, the authors should clarify exactly how data can be acquired, what is meant by ‘meet criteria for access to confidential data’, and who to contact.

The abstract is not immediately clear that per capita savings for employees actually means savings for the employer (NYC).

Several places slip from indicating PREDICTED savings to implying actual savings.

Enrollment vs enrolment spelling consistency.

The diagram is a STROBE, not CONSORT, diagram because this is an observational study.

Reviewer #4: Comments to the Author

This paper describes simulated medical cost savings due to weight loss of participants in a Weight Watchers program. The paper is motivated by trying to establish cost savings to employers by subsidizing membership fees to the Weight Watchers program. The paper is in general well written and clear to follow. That being said the analysis is based on some broad assumptions and there are some key missing features which it would be helpful to include.

Introduction –

The authors use the values of $260 on the low end and $12,548 on the high end of medical expenditure savings for weight loss taken from Crowley et al. The authors do not mention the high standard errors with these measurements or the overall analysis of the MEPS data that is conducted by Crowley et al showing significantly lower savings. Both should be addressed.

If the state is not subsidizing dependent enrollment in WW what is the states roll in providing this service to dependents?

The authors claim on line 50-52 that WW produces weight loss of approximately 5% and is cost effective. Is this an average or estimated weight loss percentage? Additionally, the reference used for the claim of cost effectiveness (29) is based on a study of prediabetic subjects not the general population. This should be addressed by the authors.

The term disease burden on line 45 is not well defined and should be explained.

Line 54 The authors state that the goal of the study is to quantify the total and per capita net annual medical expenditure savings. This appears inconsistent with the Authors’ Assumption #9.

Sample

Is the 430,000 the number of employees that meet the study’s criteria or all NYC employees?

The authors do not state why the cutoff BMI for eligibility is 25 kg/m2 or if participants with a starting BMI between 25-26 are included in cost savings or city cost calculations.

Do other dependents include children? Are medical expenditure savings the same for all age ranges?

Are there age cutoffs for the study? In Crowley et all, the estimates are for a population of adults between the ages of 24 and 64 years with biological children between the ages of 11 years (132 months) and 20 years (240 months). How does this compare to the sample in this study?

Intervention

What is behavioral skill?

Measures

Are the authors extrapolating partial year savings to full annual savings? This is not clear.

Data

The authors do not explain how they impute diabetes prevalence.

Analysis

Line 121 - The authors state that the if a participant did not have a disenrollment date they are assumed to be enrolled till the end of the evaluation period. Are there weight measurements for all these subjects at the end of the evaluation period to demonstrate that the weight loss was maintained during this time? Otherwise, what indications do the authors have that indicate weight loss besides the participants continuing to pay for their portion of program cost.

The paragraph starting on line 125 is confusing. If the evaluation period is between June 2016 and May 31, 2017 why are there weight measurements from July and August 2017? How many subjects did this occur with and were they still enrolled in the WW program at the subsidized rate from the end of May until their final weight measurements? If not, how do the authors attribute any weight loss during this time to the WW program?

Line 131 – For participants with a duration of zero, are the costs incurred by the city for these participants included in the cost savings calculation?

Line 135 – The authors state that if participants have an additional weight recorded in the three months after the evaluation period their final weight is a weighted average of the final two weights. How is this weighted average calculated and how many participants did this include? Were these participants still enrolled in the program at the subsidized rate from the end of May until their final weight measurements, if not how is weight loss after the end of the evaluation period attributed to the Weight Watchers program.

Assumptions - While briefly discussed, the assumptions the authors make to extrapolate outside published literature on cost saving due to weight loss are very broad. These assumptions should be addressed up front in the paper as they cast doubt on the validity of the cost savings analysis.

Assumption 1 - Extrapolating for cost saving for under 5% weight loss especially at lower starting BMI does not appear to be supported by reference 9 and 10.

Assumption 2 – Extrapolating cost savings for subjects under a starting BMI of 30, especial especially for weight loss of under 5 percent is a large assumption that should be addressed up front. A 5’7” male with a weight of 170lbs has a BMI of 26.62. A 2.5% reduction in weight is only 4.25lbs which is within established values for normal daily weight fluctuation.

Cost savings calculations should be reported without extrapolating below a starting BMI of 30 and without extrapolating below 5% weight loss for comparison.

Assumption 3 – The author gives no explanation for this recoding of the data.

Assumption 4 – The authors state that only participants with more than one weight recorded were eligible to contribute to savings. Did these individuals contribute to the city’s cost for subsidizing the program?

Assumption 8 – The authors appear to treat all medical expenditures as equal regarding cost. Do the authors take into account different costs for covered vs not covered procedures?

Assumption 9 – The authors assume that weight loss is immediate for all participants which is an unreasonable assumption. Additionally, the authors assume that weight loss is maintained after an individual’s final weight in. Is there any evidence that this is actually the case?

Sample Characteristics

The authors state that 91% of employees and 78% of dependents were female. Are there any differences between the cost savings due to weight loss for males vs females?

Duration

The authors state that 47.8% of employees and 45.1% of dependents did not generate cost savings. Did these participants contribute to the cost NYC incurred in the cost savings calculations?

Limitations

The authors state that self-reported weights for digital participants (42% of overall sample) were similar to other online programs that relied on measured weight. What programs are the authors using for comparison and what criteria is used to determine similarity of measurements?

The authors do not acknowledge or discuss possible increase in medical expenditures due to weight loss. i.e. An individual that needs a hip replacement but can not get the surgery until loosing weight. An individual in this situation would incur increased medical expenditures due to weight loss rather than cost savings. This should be discussed as a limitation of this analysis.

6. PLOS authors have the option to publish the peer review history of their article (what does this mean?). If published, this will include your full peer review and any attached files.

Reviewer #1: No

Reviewer #2: No

Reviewer #3: Yes: Andrew W Brown

Reviewer #4: No

---

## [Author Response · Author response to Decision Letter 0]

29 Jun 2020

Journal Requirements:

We have addressed this. 

We have addressed this. 

We have addressed this.

Due to the contractual requirements by the City of New York we are unable to share this data without a data use agreement in place. If a requestor wishes to replicate this work, they can send a data request to Eric Finkelstein at eric.finkelstein@duke-nus.edu.sg who will then initiate the process of executing the agreement. Upon approval, de-identified data will be made available to the requestor for the approved analyses.

5. Thank you for stating the following in the Financial Disclosure section: This work was supported by WW International. We note that one or more of the authors are employed by a commercial company: WW International. Please provide an amended Funding Statement declaring this commercial affiliation, as well as a statement regarding the Role of Funders in your study. If your commercial affiliation did play a role in your study, please state and explain this role within your updated Funding Statement.

Financial Disclosures: As noted in the original submission, this work was funded by WW International, a global wellness company. We now also note the following: All authors, including two from the funder, played a role in the conception and design of this study. Agrawal conducted all data analyses with input and oversight from Finkelstein. These analyses were conducted independent of the funder. Finkelstein and Agrawal led manuscript writing with input from all authors. Publication of results was agreed at the time of execution of the contract and independent of what the final results showed. 

6. Please also provide an updated Competing Interests Statement declaring this commercial affiliation along with any other relevant declarations relating to employment, consultancy, patents, products in development, or marketed products, etc. Within your Competing Interests Statement, please confirm that this commercial affiliation does not alter your adherence to all PLOS ONE policies on sharing data and materials by including the following statement: "This does not alter our adherence to PLOS ONE policies on sharing data and materials.” (as detailed online in our guide for authors http://journals.plos.org/plosone/s/competing-interests) . If this adherence statement is not accurate and there are restrictions on sharing of data and/or materials, please state these. Please note that we cannot proceed with consideration of your article until this information has been declared. Please know it is PLOS ONE policy for corresponding authors to declare, on behalf of all authors, all potential competing interests for the purposes of transparency. PLOS defines a competing interest as anything that interferes with, or could reasonably be perceived as interfering with, the full and objective presentation, peer review, editorial decision-making, or publication of research or non-research articles submitted to one of the journals. Competing interests can be financial or non-financial, professional, or personal. Competing interests can arise in relationship to an organization or another person. Please follow this link to our website for more details on competing interests: http://journals.plos.org/plosone/s/competing-interests

We have now added the following:

Competing Interests: Authors Wojtanowski, Tringali, and Foster are employees of WW International. Finkelstein is a paid consultant and paid member of the WW International Scientific Advisory Board. These competing interests do not alter our adherence to PLOS ONE policies on sharing data and materials.

7. PLOS requires an ORCID iD for the corresponding author in Editorial Manager on papers submitted after December 6th, 2016. Please ensure that you have an ORCID iD and that it is validated in Editorial Manager. To do this, go to ‘Update my Information’ (in the upper left-hand corner of the main menu), and click on the Fetch/Validate link next to the ORCID field. This will take you to the ORCID site and allow you to create a new iD or authenticate a pre-existing iD in Editorial Manager. Please see the following video for instructions on linking an ORCID iD to your Editorial Manager account: https://www.youtube.com/watch?v=_xcclfuvtxQ

This has been done as requested.

Additional Editor Comments:

The reviewers have noted concerns with the assumptions, the statistical reporting and the conclusions derived from these. The editor suggests that the results may not be extraordinary if the assumptions are revisited, however, even if they are not, reporting the results are interesting. Toning down the claims of the program as suggested by the reviewers is critical.

We have addressed all reviewers as noted below, including toning down the claims of the program. 

Reviewers' comments:

Reviewer #1: 

1) This is a well written paper to quantify the annual medical expenditure savings from an employee subsidized WW workshop or digital program weight management intervention. Overall, I believe this is a well conducted study with limitations noted in the discussion and abstract. Needed revisions are listed below.

We appreciate the positive comments from the reviewer. Specific responses to comments are noted below. 

Abstract:

2) Important to highlight that only 51% of employee and 54.1% of dependent enrollees were eligible to generate savings

We now note in the abstract: Based on the data and assumptions, 51% of employees and 54% of dependents were eligible to contribute to savings.’

3) If space allows, would also be good to highlight that the majority of savings (73% for both employees and dependents) came from enrollees with class II or III obesity.

We now note: Over 80% of savings came from participants in the Obese III category.

Methods:

4) Line 64. Note that height was also collected.

We now note in the Data Section: Participant-level data from the WW database (self-reported date of birth and height at enrollment)…. was used to conduct the analysis.

5) Line 65. Awkward sentence regarding “the perspective is NYC”. Reword.

We now note: The perspective is the New York City Government.

6) Unclear how they accounted in the ROI for employee and dependent duration of enrollment. Did the costs for individuals included differ for those enrolled four months (and employee subsidy covered 4 months) versus those enrolled six months (and employee subsidy covered 6 months)? This was not clear from the methods described.

Our approach is detailed in the main document and supporting information S1 File. In the main document we note that costs are applied monthly based on membership rates of $30 per month for Workshops and $14 per month for Digital and that NYC paid 50% of these costs for employees but did not pay for dependents. In the supplement we present the 8 step approach for quantifying savings and present two examples. In step 5 we explicitly state that we apply adjustments to savings based on participation duration (days out of 365). Operationally, we generate the savings based on a year using our assumptions and the Cawley et al estimates and then pro-rate based on enrolment duration for each participant. Note that costs accrue monthly for all participants unless they actively disenroll but savings accrue only to those with greater than 2.5% weight loss maintained for longer than 3 months. If needed we are happy to add additional details.

7) Line 163. What did the authors do for individuals with a BMI between 29.1 and 29.9?

Apologies but the subsequent sentence should have read ‘We linearly extrapolated savings between $0 and savings for a given BMI reduction for those with a starting BMI of 30 kg/m2 to impute savings for those with starting BMI values between 26 and 29.9 kg/m2. Thanks for catching this oversight.

8) Line 183-185. The authors do not provide justification for why in the sensitivity analysis, they estimate savings assuming that weight loss is maintained throughout the duration of the enrollment. Why is that assumption made given the known likelihood of weight regain once the intervention ends?

We have removed this analysis for the reasons noted. 

9) In Figure 1, the authors describe participants that were not eligible because they had an outlier for baseline height, weight, or weight change. Please add a description of what those cutoffs were, the justification for them, and how this was conducted.

This information was added to Figure 1. We now note: 4,425 participants were ineligible due to out of range age (l.t. 18yrs or g.t. 65yrs), BMI (l.t. 26kg/m2) or enrolled less than two months before the end of the evaluation period. 

We use this age range to exclude youths and because those age 65+ are Medicare eligible who is likely the primary payer of their health services. Those with a BMI below 26kg/m2 or enrolment duration of less than 3 months could not lose enough weigh to generate savings given our assumptions and thus were also excluded from the Savings (but not cost) analysis. 

We further exclude 2,134 participants from the Savings analysis because they had less than two weights recorded (so could not identify a weight change),or had outlier height, weight or weight change (beyond 99th percentile from NHANES 2011-2014 data): 

Height ≤ 58 inches or ≥ 84 inches

Weight ≤ 104 lbs. or ≥ 700 lbs.

Total weight loss ≥ 25% 

Total weight gain ≥ 25% 

Average weight loss per month ≥ 8.5%

Note that some individuals were excluded for multiple reasons. 

Results:

10) Line 247-248. This sentence is confusing.

We now note: 84% of savings came from employees, with dependents contributing 16%. 

Discussion:

11) Additional discussion is needed on the fact that only 51% of employee and 54.1% of dependent enrollees were eligible to generate savings and related limitations.

We now note in the Discussion section:

This ROI was achieved even though only 3% of employees signed up for the program and only 1/3rd of those who did lost enough weight (2.5% or more) and over a long enough period of time (3 months) to generate savings. This low uptake reflects the reality that most employees are not necessarily interested in signing up for a behavioral weight management program when offered. Moreover, many of those who signed up may have found the program not to be a good fit for them and/or were not at a stage where they were ready to commit to long term weight loss. Some of these individuals are likely to reconsider this or alternative programs at a later date. Regardless, results further reveal that, due to the high costs of obesity, even moderate weight loss among a subset of participants is enough to cover the costs of the entire subsidy and generate additional savings.

12) Additional discussion is needed on the fact that the majority of savings (73% for both employees and dependents) came from enrollees with class II or III obesity and related implications.

We now note: It is worth noting that the majority of predicted savings (84% for employees and 80% for dependents) came from participants in the Obese III category. This result is not surprising given that this group is most expensive in terms of their annual medical expenditures as shown in Cawley et al. Although this may suggest that the city (or others) should not invest in those in lower BMI categories, this is not necessarily the case. Given that individuals tend to gain weight as they age, a dynamic analysis may show that these early interventions are effective at avoiding future costs before they occur, and thus may also show a positive return on investment. This should be an area of future research.

13) There was a great deal of diversity related to employee types including administrative staff, healthcare providers, teachers, emergency services personnel and first responders, sanitation and janitorial services, among others. Do the authors have data related to job type, income, or any other indicator of socioeconomic status and how that may play a role in weight loss efforts of the employees and dependents?

Unfortunately, these data are not available to us. We agree this would make for an interesting extension and should be an area of future research. Minor comments:

There are a great deal of typos throughout the manuscript that need to be carefully checked. Some examples include:

a. Line 90. Add ‘s’ to the end of behavioral skill.

b. The word enrollment is spelled incorrectly throughout the entire manuscript (it is spelled enrolment).

c. Line 174. Comma at the end of the sentence rather than a period.

d. Line 245. Should be a space between SD and the number.

e. Line 289. Typo after analysis.

We have revisited and updated the manuscript as appropriate, including changing spelling from British to US style. 

14) Line 211. Should be clear that it is 91% of enrolled employee and dependents that were female.

We now note: 91% of employees and 78% dependents in the sample were female.

Reviewer #2

1) Main concern [Lines 157] "We assigned zero savings to those who gained weight ..." Unlike the last two items in the next section, this definitely has a structural, biasing impact on the results and is definitely unreasonable. If someone started a program and gained 10 BMI units, that's just "no savings" to the state? Not a hefty negative savings?

We appreciate the reviewer’s concern. We believe it is reasonable not to attribute weight gain to WW given that there is no evidence that WW promotes weight gain. However, it is potentially an overestimation to assume that all weight loss is attributable to WW given that some individuals may lose weight even in the absence of the program. To address both concerns, we now make it clear that our analysis considers weight gain and weight loss symmetrically, as suggested by the reviewer. 

2) [Data] If "Data cannot be shared publicly because of legal issues. Data are available from WW International for researchers who meet the criteria for access to confidential data.", then the data are not "available without restriction"

Due to the contractual requirements by the City of New York we are unable to share this data without a data use agreement in place. If a requestor wishes to replicate this work, they can send a data request to Eric Finkelstein at eric.finkelstein@duke-nus.edu.sg who will then initiate the process of executing the agreement. Upon approval, de-identified data will be made available to the requestor for the approved analyses.

3) [Line 148 and beyond] The emphasis that the Cawley estimates are "causal" (note the deliberate italics in line 148), is questionable: 1) An instrumental variable approach is just a mediational analysis with some extra assumptions; without those assumptions, there's no causality. Cawley et al. do spend time addressing/justifying this point (see, for example, the passage that mentions their references 30 and 31), but (a) reference 30 is to an entire book, which is rarely a sign of argumental strength and (b) reference 31 estimates the shared environment effect at 10%, which is not nothing

3) Lastly, to nitpick, Cawley et al. are discussing weight as an IV, not BMI. So while the argument for causality under the Cawley et al. model is stronger than the majority of other such cases where causality is invoked, this still needs some caveating by the authors.

We agree with the reviewer’s concern. To address this we have deleted the word ‘causal’ and now simply note ‘Unlike most obesity modelling studies, their modelling strategy includes an instrumental variables approach designed to address measurement error and omitted variable bias, and thus was deemed to be the best source of data for this study.’ 

4) [Lines 166-167] Item #3 in this passage seems arbitrary, unnecessary and could have a big impact on the results. Justify this choice or redo the analyses without it

Some assumptions were necessary because Cawley does not report cost savings for weight loss greater than 20% or for those with a BMI above 45kg/m2. By recoding we are making conservative assumptions that would only serve to reduce the savings. Therefore, we think this is a reasonable approach. To make this clear we now note ‘Cawley et al. also does not report savings for those with greater than 20% weight loss. Therefore, those with weight loss greater than 20% were conservatively recoded to 20%.’ and ‘For those with starting BMI values above 45 kg/m2 we conservatively recoded the BMI to 45 kg/m2 and assigned savings as reported in Cawley et al. based on the amount of weight lost.’ 

5) [Lines 236-238] Similarly, the fact that some dependents were deemed to not generate savings based on having "outlier data" is suspect.

Outliers were set based on an analysis of NHANES 2011-2014 data as follows: (a) had baseline height (<58inch or >84inch), (b) baseline weight (<104lb or >700lb), (c) weight change such that BMI reduction > 25% or BMI gain > 25% or BMI reduction per month > 8.5%. These are very conservative assumptions and only serve to reduce the ROI as these individuals still generate costs, just no savings. If there is an alternative recommendation we will update accordingly.

6) ** Lesser issues **

[Line 65] "The perspective is NYC." This statement may make sense in the authors' discipline, but here it is a non sequitur. Please make clear what it is that you are trying to convey with this passage

We now note the perspective as the New York City Government. 

7) [Line 82] "Dependents were free to also ..." Considering that this passage is talking about costs, and the costs for dependents is not zero (free), perhaps the authors should use a different phrasing

We now state ‘Dependents also had the option to…’

8) [Lines 89-90] What is meant by "discussions on behavioral skill"?

We have now expanded this section as follows: At the start of each session participants had a private check-in with a WW coach and were given the opportunity to weigh themselves, reflect on their progress, and set actionable steps for the next week. Then the weekly group discussion began with celebrating and problem solving member’s successes and challenges over the past week. After that, a new topic and technique focused on a skill related to weight loss and behavior change was introduced.

9) [Lines 100-102] It is unclear how (ii) and (iii) differ; the passages later on that describe these don't really clear this up either. Please clarify their distinction

We have clarified this such that (ii) is concerned solely with baseline weight values that extend beyond those reported in Cawley et al and (iii) is concerned with weight loss beyond what Cawley et al. reported.

10) [Lines 159-165] "for those with a BMI below 26" covers both "normal weight" and "overweight"; why is this breakpoint used?

Because we required a minimum weight loss of 2.5% and assumed no savings for those in the healthy BMI range, the minimum BMI that could generate savings is 25.7, which we conservatively rounded to 26. 

11) ** Errata **

[Line 36 and beyond] Make sure that "et al." is used, as this is short for et alia

[Line 49 and beyond] Inconsistent trademarking of Weight Watchers (compare to line 7)

[Lines 52 and beyond] Inconsistent spacing before reference numerals; see this line for a self-contained pair of examples

[Lines 99-105 and beyond] Inconsistent capitalization of items in lists; compare (i-v) and (vi) here

[Line 101 and beyond] Multiple instances of the typo: "enrolment"

[Lines 108-113] This whole paragraph is a sentence fragment

[Line 108] The parenthetical begun with "(self-reported ..." is never closed

[Line 121] Typo: "date,(s)he" (missing a space)

[Line 160] The period is part of the superscript in "m2."

[Line 174] Typo: "above,."

[Line 202 and beyond] "consort" should be "CONSORT"

We thank the reviewer for raising these issues. They have now been addressed.

Reviewer #3: 

1) The manuscript “Financial Implications of New York City's Weight Management Initiative” aimed to fill an important gap about the cost effectiveness in terms of health expenditures of widespread implementation of a commercial weight loss program, WW. Unfortunately, the model includes some fatal assumptions, some highly questionable assumptions, and inadequate handling of error.

The most substantial and fatal assumption is that individuals can only result in cost savings or no differences in costs. Individuals who have gained weight or lost less than 2.5% of weight were set to zero. However, the authors report that participants lost 6.3 to 11.6 lbs on average, but with substantial standard deviations. If weight change is assumed to be normally distributed (which the use of standard deviation would imply), then between 36% and 40% of participants would have gained weight in the sample. Claiming only success of weight losers and ignoring the costs of weight gainers necessarily biases the cost savings estimate in favor of the intervention. I note that the authors also report that 25.7% (line 232) to 27% (line 236) of participants ‘did not lose enough weight’, which indicates that the distributions are not normally distributed, and thus should not be represented by mean and SD, but still supports that a substantial proportion of the participants might have incurred cost increases rather than savings.

As noted in Response 1 to Reviewer 2, we do not believe it is unreasonable not to attribute weight gain to WW given that there is no evidence that WW promotes weight gain. However, it is potentially an overestimation to assume that all weight loss is attributable to WW given that some individuals may lose weight even in the absence of the program. To address both concerns, we now make it clear that our analysis include a more conservative approach where we consider weight gain and weight loss symmetrically, as suggested. 

2) There is also no control group, and workplace wellness program evaluations frequently note selection effects. For instance, the Illinois Workplace Wellness Study. What would these same people have experienced without WW? 

We agree that the ideal study would include a randomized trial design and a control group and now note in the limitations section ‘Future studies should aim to look at longer term cost implications of subsidizing such programs using more rigorous randomized trial designs that are free from selection and other potential biases inherent in our study’.

3) Some questionable assumptions are based on the use of the estimates from Cawley et al. Cawley et al. employed an instrumental variable approach to estimating the causal effect of weight loss on medical expenditure savings. The use of the Cawley estimates have limitations (Do they match the present sample? How does their approach to addressing self-report affect the measured vs self-report differences used in the present manuscript? Is the IV even appropriate?), but are not an altogether unreasonable basis to estimate changes in medical costs. 

Cawley estimates are generated from a nationally representative sample. We agree it’s not perfect but believe that the Cawley estimates provide the best source of data for conducting such an analysis. If the reviewer can suggest an alternative we will incorporate it. 

4) However, the present manuscript makes several questionable choices. First, rather than staying true to the original estimates, they interpolate values not reported by Cawley et al., including a 2.5% weight loss threshold that is lower than the 5% reported (it appears there is also a 7.5% threshold from tables that was not reported in the methods), and extrapolating below the BMI of 30 to include a greater n, despite Cawley et al. discussing the non-linearity of expenditures over BMI. Indeed, in Table 1 of Cawley, some of the changes in expenditure are not monotonic across % weight loss within some BMI categories. Second, the present authors chose to impute diabetes status because it was not included in the data available. This approach used a weak imputation, involving only estimating proportions based on probabilities of diabetes based on weight status alone (S1). This does not adjust for other key variables (e.g., the present population is predominantly women), and the performance of such an approach for expenditure modeling is not justified within the paper. More importantly, Table 1 of Cawley includes estimates that are averaged across all adults regardless of BMI status. The big concern is that Table 1 appears to be less favorable to support cost savings compared to Tables 2 and 3. Switching from Cawley’s model using all people to stratifying by diabetes status (without: Table 2; with: Table 3) means that cost INCREASES seen for individuals with a BMI of 30 who lose 20% of weight ($235 cost/yr) are instead estimated as SAVINGS for those without diabetes ($905 savings/yr) and with diabetes ($1865 savings/yr). This raises two concerns: why did the authors chose to use a weak imputation approach to estimate diabetes status when a combined model that did not need diabetes status was included in Cawley et al? And, if the models are that instable just from stratifying by diabetes status, are the estimates reliable for estimating cost savings in turn?

The reviewer raises two concerns related to our imputation strategy. First with respect to imputing below a BMI of 30kg/m2, because weight loss in this range has been shown to offer some health benefits (see references in manuscript), we felt that it was appropriate to consider potential savings in this range. We agree that our imputation strategy is simplistic but given the small degree of weight loss among individuals with enrollment BMI in this category, any change to the imputation strategy would have a negligible impact on the results (these individuals generate less than $10,000 in gross savings in the revised version). Re the second point, we originally attempted to control for diabetes status due to concerns that the diabetes prevalence for NYC differs from that for the general population. However, we agree that due to the weak imputation approach, this is not justified. We have now re-estimated the results using Cawley’s table 1 which combines those with and without diabetes 

5) Some procedures were not clarified: is someone who lost 4% rounded to the 5% group, rounded down to the 2.5% group, or interpolated?

We now note as Assumption 2: For each starting BMI, Cawley et al. presents savings for weight losses of 5%, 10%, 15% and 20%. For weight losses between these values, we estimated savings using linear interpolation. 

6) It appears that the error in Cawley estimates were not propagated forward into this model. That is, Person A in S1 is estimated to have experienced cost savings of $495.66, but what are the error bounds around that given the substantial error term in the Cawley tables? No estimate of uncertainty was provided for any of the model estimates, other than SD that appears based only on the point estimates from Cawley’s tables. The total estimated costs should also have an error term, and the SD reported for per capita values is only for the SD of mean estimates for each individual, rather than the variability compounded with the error presented in Cawley.

We agree that the estimates we present are associated with great uncertainty. Rather than trying to statistically convey this uncertainty, which we think would be infeasible given our inability to access the raw data in Cawley et al, we note the following in the Discussion section: Results, even when limited to employees, show an ROI of 144%. However, we caution that the actual ROI could be orders of magnitude higher or lower. We could not credibly produce confidence intervals without access to the underlying MEPS data used in Cawley et al. Regardless, results provide suggestive evidence that WW, whether delivered in person or digitally across a wide variety of employees, combined with a modest employer subsidy, has the potential to generate a positive ROI. This is an important implication that should be validated in future studies. 

7) Other error terms are confusing. For instance, given the (fatal) imposition of limiting participants to cost-savings only, the estimate of $83 per capita with an SD of 385.5 implies that 41% of people saved nothing or cost-increased (again, because SD is typically used when assuming symmetry). However, the greatest loss would be $15/mo x 12 mo = $180/yr loss. Consider presenting the data in a different format (median/IQR), or clarify how losses could be greater than $180/yr.

Because the aggregate savings and ROI are based on mean values, we believe presenting means is important. However, we agree that the presentation of SD is misleading given the skewed distribution and have now removed it in favour of presenting the median as an additional statistic. We hope the reviewer finds this change acceptable. 

8) The use of two months as a minimum length between weighings was not justified in the paper.

Evidence suggests that health improvements from weight loss can begin within weeks; therefore we believe three months is a reasonable assumption. However, to address this concern we now note in the Limitations Section: …We further assumed that weight loss translates to savings if maintained for three months or longer, however data on this assumption are limited. Validating these assumptions and results using linked weight loss and medical claims data should be an area of future research. 

9) The inclusion of weights after the end of the evaluation was also not justified. If a weight was collected during the evaluation period, it makes no immediate sense to grab weights outside of the evaluation period.

It is common in weight loss studies to impute for missing data. In our analysis, in some cases we do not have a weight at the end of the enrollment period but have the benefit of knowing an actual weight in the period just following the end of the enrollment period. We therefore take advantage of this information and use that weight and the last recorded weight to impute a weight at the end of the enrollment period as shown in Supplement S1. We see this as a conservative approach, given weights are likely to increase with time. 

10) Although I am highly critical of the assumptions above, I also want to highlight that the authors have included important adjustments: adjusting for inflation; including all costs for participants regardless of estimating cost savings (that is, if people were enrolled but did not contribute weights to be able to estimate costs); limiting cost savings to the months that experienced the weight loss (though this raises the question of whether two months would result in cost savings); limiting analysis (and participation) to those with overweight or greater BMI; adjusting costs by the % of expenditures covered by NYC benefits; and others. S1 was exceedingly helpful to walk through the complexities of the estimation procedures.

We appreciate the comments and have done our best to be responsive to the concerns raised. We are happy to make additional changes as requested such that the results fairly represent the findings of this research 

11) Other concerns:

The data availability statement does not match PLOS guidance. The answer should be ‘no’ followed by an explanation in the ‘describe where the data may be found…’ field. There, the authors should clarify exactly how data can be acquired, what is meant by ‘meet criteria for access to confidential data’, and who to contact.

Due to the contractual requirements by the City of New York we are unable to share this data without a data use agreement in place. If a requestor wishes to replicate this work, they can send a data request to Eric Finkelstein at eric.finkelstein@duke-nus.edu.sg who will then initiate the process of executing the agreement. Upon approval, de-identified data will be made available to the requestor for the approved analyses.

12) The abstract is not immediately clear that per capita savings for employees actually means savings for the employer (NYC).

This change has been made. 

13) Several places slip from indicating PREDICTED savings to implying actual savings.

Enrollment vs enrolment spelling consistency.

The diagram is a STROBE, not CONSORT, diagram because this is an observational study.

These changes have been addressed. 

Reviewer #4: 

1) This paper describes simulated medical cost savings due to weight loss of participants in a Weight Watchers program. The paper is motivated by trying to establish cost savings to employers by subsidizing membership fees to the Weight Watchers program. The paper is in general well written and clear to follow. That being said the analysis is based on some broad assumptions and there are some key missing features which it would be helpful to include.

We thank the reviewer for the positive comments and do our best to address the concerns as noted below. 

2) Introduction –

The authors use the values of $260 on the low end and $12,548 on the high end of medical expenditure savings for weight loss taken from Crowley et al. The authors do not mention the high standard errors with these measurements or the overall analysis of the MEPS data that is conducted by Crowley et al showing significantly lower savings. Both should be addressed.

In response to Review 3, we now use Cawley’s Table 1 to generate the savings without effort to impute for diabetes status. As this Table uses the full MEPS data, results are generally more precise than those reported in his Tables 2 and 3. However, results are still measured with error. Rather than trying to statistically convey this uncertainty, which we think would be infeasible given our inability to access the raw data in Cawley et al, we note the following in the Discussion section: “…It is worth noting that these results represent the best estimates of the costs, savings, and ROI to the city of New York for subsidizing the program. However, they are estimated with great uncertainty that we were unable to quantify due to the limitations above. Whether these results would generalize to other worksites should be an area of future research.”

3) If the state is not subsidizing dependent enrollment in WW what is the states roll in providing this service to dependents?

Aside from allowing dependents to enrol through the NYC portal, the city provides no role in the provision of WW but reaps the benefits of any weight losses and medical savings achieved. As we present results separately by employees and dependents, readers are free to consider only those savings they deem relevant. 

4) The authors claim on line 50-52 that WW produces weight loss of approximately 5% and is cost effective. Is this an average or estimated weight loss percentage? Additionally, the reference used for the claim of cost effectiveness (29) is based on a study of prediabetic subjects not the general population. This should be addressed by the authors.

We simplified this sentence as follows: WW provides a weight management program available in person and /or digitally. The WW program has been shown both to produce clinically meaningful weight loss [9-15] and be cost effective [16]. 

We updated the cost-effectiveness reference to include the following studies from the general population of obese adults: 

Incremental cost-effectiveness of evidence-based non-surgical weight loss strategies.

Finkelstein EA, Verghese NR. Clin Obes. 2019 Apr;9(2):e12294. doi: 10.1111/cob.12294. 

5) Line 54 The authors state that the goal of the study is to quantify the total and per capita net annual medical expenditure savings. This appears inconsistent with the Authors’ Assumption #9.

If this comment is in regard to our sensitivity analysis, where we estimate savings assuming that weight loss is maintained throughout the duration of enrolment, that assumption has been removed from the manuscript. If the author was raising another concern, we would appreciate clarification.

6) Sample

Is the 430,000 the number of employees that meet the study’s criteria or all NYC employees?

This is all NYC employees eligible to enrol based on NYC eligibility (not WW) criteria (i.e., full time staff eligible for benefits)

7) The authors do not state why the cutoff BMI for eligibility is 25 kg/m2 or if participants with a starting BMI between 25-26 are included in cost savings or city cost calculations.

As noted in response 15 to Reviewer 3, because we required a minimum weight loss of 2.5% and assumed no savings for those in the healthy BMI range, the minimum BMI that could generate savings is 25.7 kg/m2, which we rounded to 26. Therefore, those below this value were not included in the cost or savings analysis. 

8) Do other dependents include children? Are medical expenditure savings the same for all age ranges?

As noted in supporting information S1 File, analyses are limited to eligible adults between the ages of 18 and 65. The medical expenditure savings are based on the Cawley et al. results, which are not age dependent. 

9) Are there age cutoffs for the study? In Crowley et all, the estimates are for a population of adults between the ages of 24 and 64 years with biological children between the ages of 11 years (132 months) and 20 years (240 months). How does this compare to the sample in this study?

As noted above, analyses are limited to eligible adults between the ages of 18 and 65, with a mean age of 45. Although Cawley does not report a mean age, we suspect it is very close to this level given that is the midpoint of their age range.

Intervention. 

10) What is behavioral skill?

This term has been replaced with ‘behavior change strategies’. 

Measures

11) Are the authors extrapolating partial year savings to full annual savings? This is not clear.

Partial year savings are not extrapolated. If participation duration of a participant was less than a year, the predicted savings for the participant is adjusted for this duration. Therefore, a participant whose participation duration was 6 months will be allocated 50% of the annual savings estimated by Cawley et al. 

Data

12) The authors do not explain how they impute diabetes prevalence.

This imputation is no longer part of the analysis as we now use Cawley’s Table 1 to estimate savings. 

Analysis

13) Line 121 - The authors state that if a participant did not have a disenrollment date they are assumed to be enrolled till the end of the evaluation period. Are there weight measurements for all these subjects at the end of the evaluation period to demonstrate that the weight loss was maintained during this time? Otherwise, what indications do the authors have that indicate weight loss besides the participants continuing to pay for their portion of program cost.

We make this assumption to ensure we appropriate apply costs during active periods of enrollment. However, as stated in the Methods section under Assumption 6: All predicted savings are assumed to accrue immediately and terminate after the last weigh in. In other words, we do not allow time for savings to accrue nor do we allow for residual benefits. 

14) The paragraph starting on line 125 is confusing. If the evaluation period is between June 2016 and May 31, 2017 why are there weight measurements from July and August 2017? How many subjects did this occur with and were they still enrolled in the WW program at the subsidized rate from the end of May until their final weight measurements? If not, how do the authors attribute any weight loss during this time to the WW program?

Although the analysis period ends May 31, 15% of participants remain enrolled after this date. We use the weights in the subsequent two months simply to help us impute a final weight at the end of the enrolment period for those who remain enrolled as detailed in the supporting information S1 File. Note that taking advantage of these weight is conservative given that people are more likely to gain than lose weight over time. 

15) Line 131 – For participants with a duration of zero, are the costs incurred by the city for these participants included in the cost savings calculation?

Yes, these participants contribute costs but do not contribute to savings. 

16) Line 135 – The authors state that if participants have an additional weight recorded in the three months after the evaluation period their final weight is a weighted average of the final two weights. How is this weighted average calculated and how many participants did this include? 

This is shown in the example for Person A in supporting information S1 File. This included 15% (2,300 participants out of 14,946) of participants included in the analysis sample. 

17) Were these participants still enrolled in the program at the subsidized rate from the end of May until their final weight measurements, if not how is weight loss after the end of the evaluation period attributed to the Weight Watchers program.

Yes. These participants are still enrolled at the subsidized rate.

18) Assumptions - While briefly discussed, the assumptions the authors make to extrapolate outside published literature on cost saving due to weight loss are very broad. These assumptions should be addressed up front in the paper as they cast doubt on the validity of the cost savings analysis.

Rather than address this up front, we now provide a greater discussion of the assumptions in the limitations section, which we believe is the appropriate place for such a discussion but will move this up if the Editor requests. Note that we have also made several revisions to our assumptions in efforts to address reviewer concerns. 

19) Assumption 1 - Extrapolating for cost saving for under 5% weight loss especially at lower starting BMI does not appear to be supported by reference 9 and 10.

We have removed these references for the avoidance of doubt. 

20) Assumption 2 – Extrapolating cost savings for subjects under a starting BMI of 30, especially for weight loss of under 5 percent is a large assumption that should be addressed up front. A 5’7” male with a weight of 170lbs has a BMI of 26.62. A 2.5% reduction in weight is only 4.25lbs which is within established values for normal daily weight fluctuation.

It turns out that this extrapolation had a negligible impact on the results. Individuals with a starting BMI below 30 had only modest weight loss and generated less than $10,000 in gross savings out of a total of over $3m (based on the revised estimates). We believe it is worth keeping this information in the paper for readers to see the small effect that this BMI range has on the gross and net savings. 

21) Cost savings calculations should be reported without extrapolating below a starting BMI of 30 and without extrapolating below 5% weight loss for comparison.

We present the information in the tables separately by starting BMI category and by degree of weight loss so this information is included in the tables. 

22) Assumption 3 – The author gives no explanation for this recoding of the data.

We now note ‘Cawley et al. does not report savings for those with greater than 20% weight loss or BMI values above 45 kg/m2. Therefore…’

23) Assumption 4 – The authors state that only participants with more than one weight recorded were eligible to contribute to savings. Did these individuals contribute to the city’s cost for subsidizing the program?

Yes. All those who were enrolled contributed costs during the period of enrollment regardless of whether they provided weight data. 

24) Assumption 8 – The authors appear to treat all medical expenditures as equal regarding cost. Do the authors take into account different costs for covered vs not covered procedures?

We do not have access to that level of detail. As noted in Assumption 8, the City of New York’s base health plans covers 74.7% of medical expenditures referenced in the Cawley model. Therefore, for lack of a better alternative, we assumed this percentage of savings accrues to the city. 

25) Assumption 9 – The authors assume that weight loss is immediate for all participants which is an unreasonable assumption. Additionally, the authors assume that weight loss is maintained after an individual’s final weight in. Is there any evidence that this is actually the case?

As noted in Assumption 6, all predicted savings are assumed to accrue immediately and terminate after the last weigh in. We do not assume that savings are maintained beyond the final weigh in. We agree that the assumption that savings accrue immediately is overly optimistic, but we also believe that assuming residual benefits cease immediately after the last weigh in is overly pessimistic. We now state these assumptions explicitly in the limitations section: We further assumed that weight loss translates immediately to savings if maintained for three months or longer, but ceases immediately after the last weigh in. However, data on this assumption is limited. Validating these assumptions and results using linked weight loss and medical claims data should be an area of future research. 

26) Sample Characteristics

The authors state that 91% of employees and 78% of dependents were female. Are there any differences between the cost savings due to weight loss for males vs females?

This is possible but the Cawley estimates are not gender specific therefore we did not attempt to present estimates separately by gender. 

Duration

27) The authors state that 47.8% of employees and 45.1% of dependents did not generate cost savings. Did these participants contribute to the cost NYC incurred in the cost savings calculations?

Yes. Costs were generated for all participants for all periods of enrollment. 

28) Limitations

The authors state that self-reported weights for digital participants (42% of overall sample) were similar to other online programs that relied on measured weight. What programs are the authors using for comparison and what criteria is used to determine similarity of measurements?

We have removed this reference because we no longer are convinced that argument is appropriate. Self-report bias could occur regardless of whether or not results from our program are similar or different to measured results from other programs. We now simply state: Other limitations are that only self-reported weight was available for Digital participants.

29) The authors do not acknowledge or discuss possible increase in medical expenditures due to weight loss. i.e. An individual that needs a hip replacement but can not get the surgery until losing weight. An individual in this situation would incur increased medical expenditures due to weight loss rather than cost savings. This should be discussed as a limitation of this analysis.

This may occur but presumably this is built into the Cawley estimates thus we do not believe it is a limitation of our analysis. Moreover, whereas that is likely to occur after large weight losses among very high BMI individuals, such as after bariatric surgery, it is less likely to result from behavioural weight loss programs such as WW that tend to target individuals in the BMI 25 to 40 kg/m2 range.

---

## [Decision Letter · Decision Letter 1]

21 Aug 2020

PONE-D-19-29326R1

Financial implications of New York City's weight management initiative

PLOS ONE

Dear Dr. Finkelstein,

Thank you for submitting your manuscript to PLOS ONE. After careful consideration, we feel that it has merit but does not fully meet PLOS ONE’s publication criteria as it currently stands. Therefore, we invite you to submit a revised version of the manuscript that addresses the points raised during the review process.

In the editor's opinion, this manuscript will be read by the PloS One readership in detail.  In order to ensure high quality and reduce chances of misinterpretation of results, the author's need to address the statistical comments made by two of the reviewers.  The editor recommends implementing the suggestions given by Reviewers 3 and 4, especially taking heed to account for extending beyond statistical model predictions and statistical concerns such as regression to the mean.

We look forward to receiving your revised manuscript.

Kind regards,

Diana M. Thomas

Academic Editor

PLOS ONE

Reviewers' comments:

Reviewer's Responses to Questions

**Comments to the Author**

1. If the authors have adequately addressed your comments raised in a previous round of review and you feel that this manuscript is now acceptable for publication, you may indicate that here to bypass the “Comments to the Author” section, enter your conflict of interest statement in the “Confidential to Editor” section, and submit your "Accept" recommendation.

Reviewer #2: All comments have been addressed

Reviewer #3: (No Response)

Reviewer #4: (No Response)

2. Is the manuscript technically sound, and do the data support the conclusions?

Reviewer #2: (No Response)

Reviewer #3: No

Reviewer #4: Yes

3. Has the statistical analysis been performed appropriately and rigorously? 

Reviewer #2: (No Response)

Reviewer #3: No

Reviewer #4: Yes

4. Have the authors made all data underlying the findings in their manuscript fully available?

Reviewer #2: (No Response)

Reviewer #3: No

Reviewer #4: Yes

5. Is the manuscript presented in an intelligible fashion and written in standard English?

Reviewer #2: (No Response)

Reviewer #3: Yes

Reviewer #4: Yes

6. Review Comments to the Author

Reviewer #2: (No Response)

Reviewer #3: I appreciate the work the authors put into addressing the many reviewer comments. Despite my critiques, I am truly interested in seeing this analysis come to fruition, but the description of the approach is not clear enough that I could replicate it (sharing code may help), some assumptions still seem inappropriate, statistical summaries are still missing, and some artifacts/controls are not addressed.

My understanding of what the authors would like to accomplish is essentially to take an individual at baseline, map them to the estimates in Figure 1 of Cawley, then at the end of the period map them to Figure 1 again. Then, for each individual, subtract the estimated medical costs at time 1 from time 2 and get a change (which the authors refer to as ‘Savings’, even when negative). However, because Cawley et al. only report numerical values for the starting BMI in Table 1 with % changes in weight, the authors are imputing across these % change categories (5%, 10%, 15%, and 20% weight change). The authors use linear imputation to estimate between categories, and extrapolation to go between 5% and 0%, despite substantial non-linearity demonstrated in Figure 1 of Cawley, in addition to extrapolating below BMI of 30. The simplest approach would just be to contact Cawley et al. and ask for their parameter estimates, rather than complicating matters with imputation strategies. Regardless, it is unclear how the authors are implementing this approach, particularly for weight gain situations. The authors now say they use weight gain and weight loss estimates ‘symmetrically’ (in the response to reviewers) for costs and savings. ‘Symmetric’ is not necessarily correct because someone gaining 5% starting at BMI 30 would not be the same cost/savings as someone losing 5%. The extrapolation below 5% is still potentially inappropriate because Figure 1 shows that the nadir of the costs appears to be around 27-28 (difficult to estimate by eye). The assumption that savings starts at 0 therefore is not necessarily appropriate, as reinforced by the negative values in Cawley table 1 that show increased costs when participants of BMI 30 lose 20%. This negative value is ignored by the authors (e.g., Line 58 says a low of savings of $69, rather than a COST of $235). Similarly, the curvilinear nature of Figure 1 indicates that estimates of change will be over or underestimate depending on weight gain or loss for every individual, and this becomes more substantial as BMI increases. Although the authors justify using a 2.5% weight change cutoff, that is not the model they are proposing. If they assert that Cawley et al. have demonstrated continuous, causal estimates of changes in expenditure, the models are continuous over the full range, and thus should include even modest changes, rather than creating discontinuities. These discontinuities may be substantial at higher BMIs (e.g., $5k for a BMI of 45 losing 2.5%).

The authors state "We believe it is reasonable not to attribute weight gain to WW given that there is no evidence that WW promotes weight gain." This is an unreasonable assumption here. Ignoring that "absence of evidence is not evidence of absence", and that every weight loss program has some concern of rebound weight gain, my concern is not actually of attributing weight gain to any particular cause. It is a mathematical/statistical concern. If the authors were to synthesize a random sample with a mean actual weight change of zero, and select only those with randomly lower weights, and then attributed those apparent ‘weight losses’ to their program, they would get an apparent benefit of the program, even if they set all of those who had an increase in weight to zero. Thus, the weight gain must be included, and I am glad to see that it appears now to be, though again exactly how is unclear.

A bigger concern is that there is no control group. Saying that the program is cost effective indicates the program is having an effect. What proportion of individuals who selected the program would have spontaneously lost that degree of weight anyway? The authors’ model presumes all success is attributed to WW, which may be true, but remains unproven. The authors mention that 80% of ‘savings’ came from the obesity III+ category, which is the most highly variable portion of Cawley. Worse still, this is the portion of the distribution who are more likely to experience improvements by regression to the mean. Thus, any spontaneous contraction of this end of the distribution downward would give the appearance of substantial effectiveness of WW, and thereafter multiplied by medical expenditure estimates (c.f., https://pubmed.ncbi.nlm.nih.gov/31552422/). Perhaps published approaches to try to account for regression to the mean would address the issue for this manuscript at least in part?

The authors still present no estimate of variability in the estimates. The authors replace SD estimates (with their distributional problems) with medians, and the reader is still left wondering what the distribution of savings/costs are by starting/ending BMIs or changes in BMIs. The assertion that to “statistically convey this uncertainty [is] infeasible” is not necessarily true. Cawley et al. present variability estimates for each of the percent categories of weight change, which could be used for a resampling based approach for error estimates, for instance.

The instability of the authors’ approach is highlighted by the fact that the ROI from the first submission of this manuscript to this version almost doubled the estimated ROI, despite indicating several changes that were supposed to be ‘conservative’ (including saying they included costs from weight gain). How did this happen? How did employee mean savings increase by 50%, and net savings with dependents more than double?

Reviewer #4: Comments to the Author

The authors have made quite a substantial effort to address my previous comments and those of the other reviewers. Their responses are well balanced and have contributed to my understanding of the manuscript and I think that it is much improved. The authors have also made efforts to tone down their clams. There are still a few points that I feel need to be addressed.

A few specific points,

In the introduction the authors start discussing a minimal weight loss of 2.5% and use a BMI of 26 as their lower cutoff. When discussing the range of savings, the authors start the low end of the range at a BMI of 30 and weight loss of 5%. If the authors want to use weight loss of 2.5% and use a BMI of 26 as their lower cutoff they should start the savings range there.

The authors slip from indicating predicted savings to imply actual cost savings in lines 358 to 359 stating “As a result the program generated a positive return on investment for NYC.”

In assumption 3 the authors state that individuals with BMI over 45 were recoded to a BMI of 45. In assumption 4 the authors state that those with greater than 20% weight loss were recoded to 20%. The authors should report how many participants were recoded in each scenario.

On line 397 the statement “This study provides suggest evidence that the program…” should be reworded.

The quote from the paper used to respond to Reviewer 3, item 2 does not match what is in the paper.

When responding to Reviewer 3 item 6 the authors state that they caution that the actual ROI could be orders of magnitude higher or lower. This caution does not appear in the paper.

7. PLOS authors have the option to publish the peer review history of their article (what does this mean?). If published, this will include your full peer review and any attached files.

Reviewer #2: No

Reviewer #3: No

Reviewer #4: No

---

## [Author Response · Author response to Decision Letter 1]

26 Oct 2020

Reviewer Comments 

A) Reviewer # 1 – No additional comments

B) Reviewer #2 – No additional comments

C) Reviewer #3: 

1) I appreciate the work the authors put into addressing the many reviewer comments. Despite my critiques, I am truly interested in seeing this analysis come to fruition, but the description of the approach is not clear enough that I could replicate it (sharing code may help), some assumptions still seem inappropriate, statistical summaries are still missing, and some artifacts/controls are not addressed.

We are grateful that the reviewer is keen to help us further improve the manuscript to the point of publication. We recognize that our approach is not perfect, and admittedly would have done things differently if starting from scratch, but we believe the current approach, with the improvements noted below, is highly defensible and worthy of publication. 

We had thought our approach was transparent from point of initial submission both in the methods section and via our supplemental appendix. We also made efforts to further clarify our approach in the first revision. In this iteration we have made additional changes to address specific reviewer concerns, as discussed below. We also now include the stata code as an additional supporting information S2 File. STATA Code for the reviewer. We will gladly make this code available upon request or as part of the publication should the editor request. If the reviewer has additional specific recommendations for how we can further clarify our approach we are happy to incorporate those as well. 

2) My understanding of what the authors would like to accomplish is essentially to take an individual at baseline, map them to the estimates in Figure 1 of Cawley, then at the end of the period map them to Figure 1 again. Then, for each individual, subtract the estimated medical costs at time 1 from time 2 and get a change (which the authors refer to as ‘Savings’, even when negative). 

In hindsight and assuming we could get access to the underlying regression equation that generated Figure 1, this might have been our preferred approach. However, as our initial submission focused on weight loss only, and we estimated separate estimates for those with and without diabetes, our starting point was Cawley et al. Table 2 and 3 that showed predicted change in total annual medical expenditures ($US) from the instrumental variables model for those without (Table 2) and with (Table 3) type 2 diabetes. 

Based on feedback from the prior round of review we abandoned efforts to incorporate diabetes status and instead focused on Cawley et al. Table 1 which presents estimates for the full sample. What we did was put each WW participant in the correct row (row x) of Table 1 based on their starting BMI, then the correct column (column y) based on how much weight (BMI %) they lost and then pulled the savings from that cell (x,y) as the annual savings. We further adjust the savings based on the duration of the weight loss (i.e., if the duration is six months we give them half of Cawley et al. annual savings). 

3) However, because Cawley et al. only report numerical values for the starting BMI in Table 1 with % changes in weight, the authors are imputing across these % change categories (5%, 10%, 15%, and 20% weight change). The authors use linear imputation to estimate between categories, and extrapolation to go between 5% and 0%, despite substantial non-linearity demonstrated in Figure 1 of Cawley, in addition to extrapolating below BMI of 30. The simplest approach would just be to contact Cawley et al. and ask for their parameter estimates, rather than complicating matters with imputation strategies. 

We agree that in hindsight contacting Cawley and asking for the parameter estimates might have been preferred, but we believe our imputation/extrapolation approach is completely defensible. Below we reproduce Cawley et al. Figure 1, but add the dark blue line segments between each 5 unit BMI value. What can be seen from the Figure is that despite substantial non-linearities across the entire BMI spectrum as the reviewer notes, BMI is largely linear within each 5 BMI unit line segment between 25 and 40, thus justifying our imputation approach. In fact, only between BMI 35 and 40 do we clearly see the non-linearity coming through and given that the curve falls below our line segment, it means that our assumption of linearity within this segment is conservative (we assign slightly lower savings for a reduction from, for example, BMI 40 to 37 then Cawley et al. curve would predict). For other ranges his curve and our line overlap. For these reasons, we believe our approach for estimating savings is reasonable. 

 

4) Regardless, it is unclear how the authors are implementing this approach, particularly for weight gain situations. The authors now say they use weight gain and weight loss estimates ‘symmetrically’ (in the response to reviewers) for costs and savings. ‘Symmetric’ is not necessarily correct because someone gaining 5% starting at BMI 30 would not be the same cost/savings as someone losing 5%. 

We appreciate the reviewers comment and have clarified our approach. We now under Assumption 1: 

Cawley et al. only reported savings for those with weight loss of 5% or greater. We linearly extrapolated savings for all those with BMI reduction (i.e., weight loss) between 2.5% and 5%. For example, if an individual had a 2.5% reduction in BMI then she would receive medical cost savings of half of the corresponding value in the 5% BMI reduction column in Cawley et al. Table 1. Clinical evidence suggests that weight loss as little as 2.5% generates clinical health improvements [4-6] thus, we deemed it reasonable to apply cost savings for weight losses of this magnitude or greater, although we explore the implications of this assumption in the sensitivity analysis. For weight losses smaller than 2.5% no savings were assumed. 

And under Assumption 6:

To minimize risks of bias, we apply the estimates in Cawley et al. Table 1 in reverse for those who gained weight. For example, someone with a starting BMI of 33kg/m2 who lost 5% of their baseline weight would have a predicted savings in annual medical expenditures of $288 (in $2010). If a different individual with the same starting BMI gained 5% of their baseline weight we apply the $288 as a cost, not as a savings. This conservative approach addresses concerns due to omitted variables, mean reversion [20], and other potential sources of bias. 

We further note in the limitations section: 

Although we conservatively applied costs for those who gained weight, we assumed that BMI increases of a given percentage have the same effect on annual medical expenditures, in absolute terms, as BMI reductions. In reality, risks of chronic disease and medical expenditures increase non-linearly with increasing BMI. Therefore, our cost increases for those who gained weight may be underestimated. This is further exacerbated by applying zero costs for overweight individuals who gained weight. However, given only 15% of our sample, including overweight participants at baseline, gained weight and average weight gain was only 5.7lbs, varying these assumptions is likely to have little impact on the resulting ROI estimate.

5) The extrapolation below 5% is still potentially inappropriate because Figure 1 shows that the nadir of the costs appears to be around 27-28 (difficult to estimate by eye). The assumption that savings starts at 0 therefore is not necessarily appropriate, as reinforced by the negative values in Cawley table 1 that show increased costs when participants of BMI 30 lose 20%. This negative value is ignored by the authors (e.g., Line 58 says a low of savings of $69, rather than a COST of $235). Similarly, the curvilinear nature of Figure 1 indicates that estimates of change will be over or underestimate depending on weight gain or loss for every individual, and this becomes more substantial as BMI increases. Although the authors justify using a 2.5% weight change cutoff, that is not the model they are proposing. If they assert that Cawley et al. have demonstrated continuous, causal estimates of changes in expenditure, the models are continuous over the full range, and thus should include even modest changes, rather than creating discontinuities. These discontinuities may be substantial at higher BMIs (e.g., $5k for a BMI of 45 losing 2.5%).

Some of the issues the reviewer raises here were addressed in the prior responses. However, two issues remain, which we address in turn. 

Extrapolating weight losses between 0 and 2.5%:

Even though Cawley’s continuous model would allow for estimating savings for weight change below 2.5%, we do not believe the clinical evidence provides justification for such an assumption. Therefore, we believe it is appropriate to assume zero savings (or costs) for weight change below this percentage. We note in the manuscript: Clinical evidence suggests that weight loss as little as 2.5% generates clinical health improvements [4-6] thus, we deemed it reasonable to apply cost savings for weight losses of this magnitude or greater.

Extrapolating weight losses below 30 BMI. 

This is another excellent point raised by the reviewer and one we did not adequately consider in prior versions. Cawley et al. state in the manuscript that the nadir on costs occurs at a BMI of 25 kg/m2, but the reviewer rightly points out that expenditures appear unaffected by changes in BMIs within the overweight range and that weight losses that generate BMI values below 25kg/m2 may lead to weight gain. 

Given our current methods, we believe the most straightforward and fully defensible approach is to apply zero savings from weight loss to anyone whose starting BMI is below 30 kg/m2(i.e., not extrapolating to BMI values below those listed in Cawley et al. Table 1). We now note under Assumption 3: We assumed $0 savings for those with a starting BMI in the overweight range due to lack of evidence that weight loss translates into reductions in medical expenditures for this BMI category.

We also note in the limitations section, as stated above ‘... Therefore, our cost increases for those who gained weight may be underestimated. This is further exacerbated by applying zero costs for overweight individuals who gained weight. However, given only 15% of our sample, including overweight participants at baseline, gained weight and average weight gain was only 5.7lbs, varying these assumptions is likely to have little impact on the resulting ROI estimate.

6) The authors state "We believe it is reasonable not to attribute weight gain to WW given that there is no evidence that WW promotes weight gain." This is an unreasonable assumption here. Ignoring that "absence of evidence is not evidence of absence", and that every weight loss program has some concern of rebound weight gain, my concern is not actually of attributing weight gain to any particular cause. It is a mathematical/statistical concern. If the authors were to synthesize a random sample with a mean actual weight change of zero, and select only those with randomly lower weights, and then attributed those apparent ‘weight losses’ to their program, they would get an apparent benefit of the program, even if they set all of those who had an increase in weight to zero. Thus, the weight gain must be included, and I am glad to see that it appears now to be, though again exactly how is unclear.

We appreciate the reviewer’s concern and have now included medical costs for weight gain and provided additional clarity for our approach as described in response 4 above.

7) A bigger concern is that there is no control group. Saying that the program is cost effective indicates the program is having an effect. 

We appreciate the authors concern but note that the evidence for the effectiveness of WW is already well established via high quality randomized trials, as we cite in the introduction. That is not the focus of this analysis. Rather our goal is to use real world evidence to simulate the potential savings from this evidence based program. We agree that the ideal study would have a control group and make this point clear in the manuscript, however, we still maintain that this manuscript provides important information, despite the limitations. 

8) What proportion of individuals who selected the program would have spontaneously lost that degree of weight anyway? The authors’ model presumes all success is attributed to WW, which may be true, but remains unproven. 

We appreciate the reviewer’s concern but maintain that results of real world evidence studies are important to publish despite their limitations. However, we agree that these studies need to be transparent and limitations noted. We now state in the limitations section: Importantly, although these results represent the best estimates of the costs, savings, and ROI associated with the subsidized program, we cannot say the program caused the savings. This results, because, in the absence of the subsidy, some employees would have made efforts to lose weight either by joining WW and paying on their own, or in myriad other ways, and some would have been successful. Therefore, these results may represent an upper bound of the potential savings resulting from the subsidized weight management program. …. Future studies could improve on these estimates by using linked data on weight change and changes in medical expenditures using more rigorous randomized trial designs that are free from selection and the other potential biases and that can incorporate both parametric and non-parametric sensitivity analyses….

9) The authors mention that 80% of ‘savings’ came from the obesity III+ category, which is the most highly variable portion of Cawley. Worse still, this is the portion of the distribution who are more likely to experience improvements by regression to the mean. Thus, any spontaneous contraction of this end of the distribution downward would give the appearance of substantial effectiveness of WW, and thereafter multiplied by medical expenditure estimates (c.f., https://pubmed.ncbi.nlm.nih.gov/31552422/). Perhaps published approaches to try to account for regression to the mean would address the issue for this manuscript at least in part?

We address variability in our response to Comment 11 below but remain unconvinced that spontaneous contraction or regression to the mean is a significant concern here. The referenced manuscript suggests that misattribution of RTM typically occurs when investigators restrict analyses in a segment of the sample above or below the population mean to determine pretest to posttest intervention changes and do not utilize a control group. In our case, we include all those with BMI values above 30 kg/m2 at baseline so have not restricted the sample to the extreme ends of the distribution (at least not for NYC). It is true that most of our savings comes from the Obese III group, but this is to be expected as this is the group that has the highest health risks at baseline and therefore the most potential for savings due to weight loss. Using the authors example of the PEACH child overweight study, one way to test for RTM is via longitudinal data of ‘controls’. With respect to Obese III adults, longitudinal data reveals that, in the absence of an intervention, they do not tend to lose weight or ‘regress to the mean’. In fact, as shown in the Figure below, reproduced from Young Adult Weight Trajectories Through Midlife by Body Mass Category Rahul Malhotra 1, Truls Ostbye, Crystal M Riley, Eric A Finkelstein, Obesity (Silver Spring). 2013 Sep;21(9):1923-34. doi: 10.1002/oby.20318., 68% of those who enter adulthood with a BMI in the Obese III range will continue to gain weight as they age at least through midlife. Moreover, even for the 32% who lose weight, average annual weight loss is small, less than one pound per year. As such, whereas we agree that our analysis is subject to several potential biases due to lack of a control group, something we readily acknowledge and address in the manuscript, we do not believe additional efforts are needed to specifically address RTM concerns. 

10) The authors still present no estimate of variability in the estimates. The authors replace SD estimates (with their distributional problems) with medians, and the reader is still left wondering what the distribution of savings/costs are by starting/ending BMIs or changes in BMIs. 

We address the variability issue in response to Comment 11. In the prior submission we presented Gross savings by Enrolment BMI (Table 3) and by BMI reduction (Table 4). We now present, in the supporting information S1 Table, net savings for these same categories. As discussed in Comment 11, we also present Gross and Net Savings and ROI for a series of sensitivity analyses in Table 6. We hope this addresses the reviewer concerns but will include additional analyses should the reviewer or editor request. 

11) The assertion that to “statistically convey this uncertainty [is] infeasible” is not necessarily true. Cawley et al. present variability estimates for each of the percent categories of weight change, which could be used for a resampling based approach for error estimates, for instance.

The reviewer is correct that a statistical approach to uncertainty may be feasible. However, we propose a non-parametric approach that is common in economic evaluations. We present several one-way sensitivity analyses that vary key parameters of interest. For this analysis, these parameters include 1) minimum weight loss to generate cost savings, 2) minimum duration that weight loss must be maintained, and 3) costs savings for any given starting BMI/weight loss. Therefore, in addition to our primary estimates, we now include gross and net savings and ROI under the following scenarios:

1) Minimum weight loss to generate expenditure savings is 5% (as opposed to 2.5%)

2) Minimum duration to generate expenditure savings is 6 months (as opposed to 3 months, and

3) Costs are 50% of what is reported in Cawley et al. Table 1. 

We believe this approach will give the reader a sense of the variability in the estimates and addresses the reviewers concerns. 

In the limitations section, we also note:…These estimates are also associated with great uncertainty. In efforts to gauge the influence of key parameters on the estimates, we present savings and ROI estimates both by applying stratifications that allow for determining which population subsets are most influential in the ROI calculation and by including one-way sensitivity analyses of key input parameters. Even in our most conservative scenario of savings reduced by 50%, the forecast ROI remained positive. Future studies could improve on these estimates by using linked data on weight change and changes in medical expenditures using more rigorous randomized trial designs that are free from selection and the other potential biases and that can incorporate both parametric and non-parametric sensitivity analyses…

12) The instability of the authors’ approach is highlighted by the fact that the ROI from the first submission of this manuscript to this version almost doubled the estimated ROI, despite indicating several changes that were supposed to be ‘conservative’ (including saying they included costs from weight gain). How did this happen? How did employee mean savings increase by 50%, and net savings with dependents more than double?

The reviewer may recall that the first submission relied on a weighted average of savings presented in Cawley et al. Table 2 (non-diabetics) and Table 3 (diabetics), with the weights representing the probability of the participant having diabetes. Based on the reviewer feedback, the analysis in all subsequent versions is based on estimates from Cawley et al. Table 1, which relies on the full underlying MEPS data. Because Cawley’s modelling approach is non-linear, this had a large impact on the estimates. Let me give an example to demonstrate how this change is driving the change in the savings estimates. 

Example: Participant enrollment BMI = 40 kg/m2, assumed probability of diabetes from NYC HANES 2013-14 data set for participant with enrollment BMI = 40 kg/m2 is 0.296. BMI reduction= 5%.

From Cawley et al. Table 1, the estimated medical saving for participant with enrollment BMI 40kg/m2 and a 5% BMI reduction is $2137.15 ($2010). From Cawley et al. Table 2, the estimated medical saving for participant with enrollment BMI 40kg/m2 and a 5% BMI reduction for participant without diabetes is $643.39 ($2010). From Cawley et al. Table 3, the estimated medical saving for participant with enrollment BMI 40kg/m2 and a 5% BMI reduction for participant with diabetes is $2122.72 ($2010). Therefore, the weighted average medical saving from Table 2 and Table 3 estimates = 0.296*2122.73 + (1-0.296)*643.39 = $1081.27. This is almost half the savings estimated from Table 1 for the same participant. 

Therefore, Cawley et al. Table 1 estimates of savings are driving the higher savings estimates than the weighted average of savings from Cawley et al. Table 2 and Table 3. 

We acknowledge that our estimates are associated with great uncertainty but believe this is adequately address through our transparent approach, inclusion of sensitivity analyses with costs at 50% of Cawley’s base case values, and via raising the concern in the limitations section as noted above. We hope you agree this is sufficient to let readers decide on the veracity of our approach and resulting estimates.

D) Reviewer #4: Comments to the Author

1) The authors have made quite a substantial effort to address my previous comments and those of the other reviewers. Their responses are well balanced and have contributed to my understanding of the manuscript and I think that it is much improved. The authors have also made efforts to tone down their clams. There are still a few points that I feel need to be addressed.

We appreciate the positive comments and address the remaining points as noted below. 

2) In the introduction the authors start discussing a minimal weight loss of 2.5% and use a BMI of 26 as their lower cutoff. When discussing the range of savings, the authors start the low end of the range at a BMI of 30 and weight loss of 5%. If the authors want to use weight loss of 2.5% and use a BMI of 26 as their lower cutoff they should start the savings range there.

Thanks for the comment. Cawley et al. paper did not consider those with less than 5% weight loss or those with BMI’s below 30 kg/m2. That is why we had to stick with these values. To clarify, we now note: Cawley et al [8] used nationally representative data and econometric analyses to estimate medical costs savings resulting from weight loss of 5% or greater for those with BMI values ranging between 30 and 45 kg/m2. He reports that, even with just 5% weight loss, the estimated annual savings in medical costs are $US69 (in $2010) for those with a starting BMI of 30 kg/m2, $US528 for those with a starting BMI of 35 kg/m2, and $US2,137 for those with a starting BMI of 40 kg/m2. In most cases savings are predicted to increase with greater weight loss.

3) The authors slip from indicating predicted savings to imply actual cost savings in lines 358 to 359 stating “As a result the program generated a positive return on investment for NYC.” 

We agree. This sentence was not needed and has been removed. 

4) In assumption 3 the authors state that individuals with BMI over 45 were recoded to a BMI of 45. In assumption 4 the authors state that those with greater than 20% weight loss were recoded to 20%. The authors should report how many participants were recoded in each scenario. 

1. We now note: For the less than 1% of participants with starting BMI values above 45 kg/m2 (n=858) we conservatively recoded the BMI to 45 kg/m2 and assigned savings as reported in Cawley et al. based on the amount of weight lost.

2. …. Cawley et al. also do not report savings for those with greater than 20% weight loss. Therefore, for those few cases with weight change greater than 20% (n=58 for weight loss, n=7 for weight gain), we recoded the change to 20%.

5) On line 397 the statement “This study provides suggest evidence that the program…” should be reworded.

We have reworded this sentence to state: This study shows the potential for modest subsidies of evidence based, commercially available weight management programs, whether delivered in person or digitally, to generate a positive return on investment. 

6) The quote from the paper used to respond to Reviewer 3, item 2 does not match what is in the paper. 

We apologize for this oversight. We have ensured that the reviewer responses and manuscript are now in concert. 

7) When responding to Reviewer 3 item 6 the authors state that they caution that the actual ROI could be orders of magnitude higher or lower. This caution does not appear in the paper.

Thanks for raising this point. We have addressed Reviewer 3’s concerns as noted Response 8 to Reviewer 3.

---

## [Decision Letter · Decision Letter 2]

4 Jan 2021

PONE-D-19-29326R2

Financial implications of New York City's weight management initiative

PLOS ONE

Dear Dr. Finkelstein,

Thank you for submitting your manuscript to PLOS ONE. After careful consideration, we feel that it has merit but does not fully meet PLOS ONE’s publication criteria as it currently stands. Therefore, we invite you to submit a revised version of the manuscript that addresses the points raised during the review process.

The authors have been patient and transformative with this manuscript.  At this stage, both reviewers have identified minor typos and consistency errors which should be trivial to correct.  The authors should address these comments and hopefully this will not take long.

We look forward to receiving your revised manuscript.

Kind regards,

Diana M. Thomas

Academic Editor

PLOS ONE

Reviewers' comments:

Reviewer's Responses to Questions

**Comments to the Author**

1. If the authors have adequately addressed your comments raised in a previous round of review and you feel that this manuscript is now acceptable for publication, you may indicate that here to bypass the “Comments to the Author” section, enter your conflict of interest statement in the “Confidential to Editor” section, and submit your "Accept" recommendation.

Reviewer #3: All comments have been addressed

Reviewer #4: (No Response)

2. Is the manuscript technically sound, and do the data support the conclusions?

Reviewer #3: Yes

Reviewer #4: Yes

3. Has the statistical analysis been performed appropriately and rigorously? 

Reviewer #3: Yes

Reviewer #4: Yes

4. Have the authors made all data underlying the findings in their manuscript fully available?

Reviewer #3: No

Reviewer #4: No

5. Is the manuscript presented in an intelligible fashion and written in standard English?

Reviewer #3: Yes

Reviewer #4: Yes

6. Review Comments to the Author

Reviewer #3: I thank the authors for addressing my many comments. This revision clarified multiple aspects of the study and while we could argue about assumptions ad nauseum, the authors have made their assumptions and processes clear to the reader. The clarity of assumptions, including the sensitivity analyses, give the reader an idea of the ramifications of various assumptions, which is critical for a paper of this sort. Thank you.

I few minor points the authors should consider, or would hopefully be caught in proofing:

-The Roman numerals in the Measures section repeats iii; all subsequent ones should be renumbered.

-Line 299: a distracted (or nefarious) reader may misinterpret the phrase 'based on the mean values' to be selecting means over medians to inflate the total estimates, whereas the summaries are just sums of savings (equivalent to multiplying means by n). I recommend removing "Based on the mean values".

-Tables may benefit from legends/footnotes so they can better stand on their own for interpretability.

-Line 434: perhaps rephrase "we do not see this as a concern" as it might read as though the authors are not concerned about the people mentioned.

-Line 452+ regarding addressing 'great uncertainty': I appreciate the different models. Perhaps in the methods and here the authors could briefly but explicitly make it clear that rather than focusing on estimating point-estimate error bounds based on error assumptions (e.g., using the variability in Cawley) the authors chose to provide sensitivity analyses with very different model assumptions. Just in case a reader is wondering where error bars/estimates are, this would address it clearly.

-Figure 1: the exclusion says BMI<26, but the authors indicate in the methods that exclusion was BMI<=25

Reviewer #4: Comments to the Author

I commend the authors as they have made quite a substantial effort to address my previous comments and those of the other reviewer. Their responses are thorough and enhance the reader’s understanding of their methods. The authors have also made additional efforts to tone down their clams and discuss the limitations of their study. I appreciate the sensitivity analysis as I think it allows the reader to assess the impact of the author’s assumptions and gives more validity to the work. There are just a couple points that I feel need to be addressed.

A couple of specific points,

The authors have done well to state that their savings calculations are predicted and not actually realized savings. However, on line 137 “Annual net saving for NYC and ROI” are listed as calculated measures. It should be made clear that both values are predictions.

On line 204-205 the authors state that “For the less than 1% of participants with starting BMI values above 45 (n=858) we conservatively recoded the BMI to 45.” The n of 858 and less than 1% of participants does not appear to match. I am assuming one of these numbers is an error.

On line 252 the authors state that costs are 50% of the Cawley et al. Table 1. Since the values in Table 1 are reductions in cost I believe the authors should refer to these values as savings rather than costs.

The value of gross predicted savings to NYC of $2,999,858 does not account for the increased cost of weight gain and does not match the total gross savings listed in tables 2, 3, and 4 of $2,522,529. The authors also use this value of $2,999,858 for total gross savings in the base case in their sensitivity analysis. It would be more appropriate to use $2,522,529 as the base case in this analysis.

On line 399-400 the authors state that “This results because, among the 19% who generated savings, savings were large.” It is unclear where the 19% comes from as table 4 shows that 35% of participants had weight loss that attributed to savings.

7. PLOS authors have the option to publish the peer review history of their article (what does this mean?). If published, this will include your full peer review and any attached files.

Reviewer #3: No

Reviewer #4: No

---

## [Author Response · Author response to Decision Letter 2]

11 Jan 2021

Reviewer #3: 

I thank the authors for addressing my many comments. This revision clarified multiple aspects of the study and while we could argue about assumptions ad nauseum, the authors have made their assumptions and processes clear to the reader. The clarity of assumptions, including the sensitivity analyses, give the reader an idea of the ramifications of various assumptions, which is critical for a paper of this sort. Thank you.

We thank the reviewer for the positive comment.

I few minor points the authors should consider, or would hopefully be caught in proofing:

-The Roman numerals in the Measures section repeats iii; all subsequent ones should be renumbered.

Thank you for pointing this out. We have now renumbered the Roman numerals in the ‘Measures’ section to rectify this mistake. 

-Line 299: a distracted (or nefarious) reader may misinterpret the phrase 'based on the mean values' to be selecting means over medians to inflate the total estimates, whereas the summaries are just sums of savings (equivalent to multiplying means by n). I recommend removing "Based on the mean values".

We agree that the wording can be misinterpreted. We have removed the words “Based on the mean values” and the sentence now reads “Total predicted savings for employees was $1,940,050 and $582,479 for ‘Workshops’ and ‘Digital’ respectively.”

-Tables may benefit from legends/footnotes so they can better stand on their own for interpretability.

We believe the tables are clear as presented but will add additional footnotes if the editor requests. 

-Line 434: perhaps rephrase "we do not see this as a concern" as it might read as though the authors are not concerned about the people mentioned.

Thank you for pointing this out. We have removed this part of the sentence and simply note that: “However, for those with starting BMI values above 30 kg/m2 in our data, only 0.05% lost enough weight such that their final BMI was below 25 kg/m2 at follow up.”

-Line 452+ regarding addressing 'great uncertainty': I appreciate the different models. Perhaps in the methods and here the authors could briefly but explicitly make it clear that rather than focusing on estimating point-estimate error bounds based on error assumptions (e.g., using the variability in Cawley) the authors chose to provide sensitivity analyses with very different model assumptions. Just in case a reader is wondering where error bars/estimates are, this would address it clearly.

Due to word count restrictions we address this concern solely in the methods section. We now note the following on line 243: In efforts to gauge the sensitivity of the ROI results to key model assumptions, and given the inability to access the raw data presented in Cawley et al for producing confidence intervals around the estimates, we present a series of one-way sensitivity analyses. The key assumptions are…

-Figure 1: the exclusion says BMI<26, but the authors indicate in the methods that exclusion was BMI<=25

Thank you for pointing this out and we apologize for the oversight. We now fix line 100 to read: “Individuals who are employed by one of the 150 collective bargaining units and eligible for health benefits under the NYC Health Benefits Program and with a BMI of 26 kg/m2 or more were eligible for WW program enrollment.”

Reviewer #4: Comments to the AuthorI commend the authors as they have made quite a substantial effort to address my previous comments and those of the other reviewer. Their responses are thorough and enhance the reader’s understanding of their methods. The authors have also made additional efforts to tone down their clams and discuss the limitations of their study. I appreciate the sensitivity analysis as I think it allows the reader to assess the impact of the author’s assumptions and gives more validity to the work. 

Thank you for the positive comments. We greatly appreciate it. 

There are just a couple points that I feel need to be addressed. A couple of specific points,

1) The authors have done well to state that their savings calculations are predicted and not actually realized savings. However, on line 137 “Annual net saving for NYC and ROI” are listed as calculated measures. It should be made clear that both values are predictions. 

Thank you for pointing this out. We have changed the text to “(vi) Predicted annual net savings for NYC and predicted ROI.”

2) On line 204-205 the authors state that “For the less than 1% of participants with starting BMI values above 45 (n=858) we conservatively recoded the BMI to 45.” The n of 858 and less than 1% of participants does not appear to match. I am assuming one of these numbers is an error. 

Thank you for pointing this out. 858 participants is correct but our math was wrong. The sentence now reads: “For the 5.7% of participants with starting BMI values above 45 kg/m2 (n=858) we conservatively recoded their BMI to 45 kg/m2 and assigned savings as reported in Cawley et al. based on the amount of weight lost.”

3) On line 252 the authors state that costs are 50% of the Cawley et al. Table 1. Since the values in Table 1 are reductions in cost I believe the authors should refer to these values as savings rather than costs.

Thanks for pointing this out. We now note:“Medical expenditure savings are 50% of what is reported in Cawley et al. Table 1.” 

4) The value of gross predicted savings to NYC of $2,999,858 does not account for the increased cost of weight gain and does not match the total gross savings listed in tables 2, 3, and 4 of $2,522,529. The authors also use this value of $2,999,858 for total gross savings in the base case in their sensitivity analysis. It would be more appropriate to use $2,522,529 as the base case in this analysis. 

The value of gross predicted savings to NYC of $2,999,858 is the total gross predicted savings for employees and dependents combined and it does take into account the increased cost of weight gain. If you look at Tables 2,3, and 4, the gross predicted savings are separately listed as $2,522,529 for employees and $477,329 for dependents. These sums up to a total gross predicted savings to NYC of $2,999,858. Hope this clarifies why we use total gross predicted savings of $2,999,858 as the base case in this analysis. 

5) On line 399-400 the authors state that “This results because, among the 19% who generated savings, savings were large.” It is unclear where the 19% comes from as table 4 shows that 35% of participants had weight loss that attributed to savings. 

We agree this point was not clear. We have now added the following at line 347 to clarify: “Table 4 consists of those eligible to generate savings. However, even if weight loss exceeds 2.5%, eligible participants may not generate savings. This would occur if they do not meet other criteria, including participation duration less than 90 days and/or enrollment BMI less than 30 kg/m2. Although Table 4 shows that 35% of participants had weight losses of 2.5% or more, only 2,426 participants (19%) out of those eligible to contribute to savings actually did so.” Hope this is now clear. Thanks again for the feedback.

---

## [Editor Report · Decision Letter 3]

25 Jan 2021

Financial implications of New York City's weight management initiative

PONE-D-19-29326R3

Dear Dr. Finkelstein,

We’re pleased to inform you that your manuscript has been judged scientifically suitable for publication and will be formally accepted for publication once it meets all outstanding technical requirements.

Kind regards,

Diana M. Thomas

Academic Editor

PLOS ONE
---

## [Editor Report · Acceptance letter]

28 Jan 2021

PONE-D-19-29326R3 

Financial implications of New York City’s weight management initiative 

Dear Dr. Finkelstein:

I'm pleased to inform you that your manuscript has been deemed suitable for publication in PLOS ONE. Congratulations! Your manuscript is now with our production department. 

Kind regards, 

on behalf of

Dr. Diana M. Thomas 

Academic Editor

PLOS ONE